# Pressure dependence in aqueous-based electrochemical CO$_2$ reduction

Liang Huang [1,2,7], Ge Gao[1,2,7], Chaobo Yang[1,3,7], Xiao-Yan Li[4,7], Rui Kai Miao [5,7], Yanrong Xue[1,2], Ke Xie[4], Pengfei Ou [4], Cafer T. Yavuz [6], Yu Han [6], Gaetano Magnotti[1] ✉, David Sinton [5] ✉, Edward H. Sargent [4] ✉ & Xu Lu [1,2] ✉

Electrochemical CO$_2$ reduction (CO$_2$R) is an approach to closing the carbon cycle for chemical synthesis. To date, the field has focused on the electrolysis of ambient pressure CO$_2$. However, industrial CO$_2$ is pressurized—in capture, transport and storage—and is often in dissolved form. Here, we find that pressurization to 50 bar steers CO$_2$R pathways toward formate, something seen across widely-employed CO$_2$R catalysts. By developing *operando* methods compatible with high pressures, including quantitative *operando* Raman spectroscopy, we link the high formate selectivity to increased CO$_2$ coverage on the cathode surface. The interplay of theory and experiments validates the mechanism, and guides us to functionalize the surface of a Cu cathode with a proton-resistant layer to further the pressure-mediated selectivity effect. This work illustrates the value of industrial CO$_2$ sources as the starting feedstock for sustainable chemical synthesis.

Electrochemical CO$_2$ reduction (CO$_2$R) to chemicals offers one form of upgrading/utilizing captured CO$_2$[1–3]. CO$_2$R has demonstrated a wide range of products, including carbon monoxide (CO), formate/formic acid, alcohols, and hydrocarbons, at industrially relevant reaction rates[4–6]. When powered using renewable electricity, CO$_2$R can reduce the carbon intensity of the otherwise fossil fuel-based production of these carbon compounds[7–10]. To date, most studies have focused on electrolysis of ambient pressure CO$_2$[11,12]. In actuality, many processes involving CO$_2$ work at pressure (PCO$_2$), with CO$_2$ typically in dissolved form (1–110 bar)[13]. PCO$_2$ is the effluent of industrial processes such as natural gas reforming and ethylene oxide production (3–20 bar)[14,15]. Depressurizing these CO$_2$ sources to accommodate existing ambient pressure CO$_2$R cells incurs an energy penalty and unnecessarily takes downhill the total energetic value of the reactant.

PCO$_2$ also benefits from high CO$_2$ solubility in aqueous solutions. At ambient pressures, CO$_2$'s low solubility diminishes current densities to the vicinity of few ~10 milliamperes per square centimeter[16,17]. Dissolving PCO$_2$ in liquid electrolyte delivers much more reactant to the CO$_2$R catalyst: the CO$_2$ concentration increases from 0.03 M under ambient pressure to 1.16 M under 50 bar[18]. In addition, adopting dissolved PCO$_2$ during CO$_2$R could stabilize the bulk catholyte pH at ~6.2, which is otherwise alkalized amid stoichiometric OH$^-$ production[19].

In prior studies that sought to lever pressure in aqueous-based CO$_2$R[20,21], PCO$_2$ was reduced to CO or formate[22]. These include examinations of altered CO$_2$R product selectivity on various metal catalysts under high pressure[23–26]. A Ni wire electrode that had no CO$_2$R activity under ambient pressure showed 23% formic acid selectivity under 60 bar[25]. Enhanced formate selectivity was seen on Sn using PCO$_2$[27,28]. Theoretical modeling and control experiments were also conducted to understand CO$_2$R under high pressure[29–33]. More recently, PCO$_2$ has been found to transform Cu-based catalysts to become formate-selective[34].

[1]CCRC, Division of Physical Science and Engineering (PSE), King Abdullah University of Science and Technology (KAUST), Thuwal 23955-6900, Saudi Arabia. [2]KAUST Solar Center (KSC), PSE, KAUST, Thuwal 23955-6900, Saudi Arabia. [3]National Key Laboratory of Science and Technology on Tunable Laser, Harbin Institute of Technology, Harbin 150001, China. [4]Department of Electrical and Computer Engineering, University of Toronto, 10 King's College Road, Toronto, Ontario M5S 3G4, Canada. [5]Department of Mechanical and Industrial Engineering, University of Toronto, 5 King's College Road, Toronto, Ontario M5S 3G8, Canada. [6]Advanced Membranes and Porous Materials Center (AMPM), PSE, KAUST, Thuwal 23955-6900, Saudi Arabia. [7]These authors contributed equally: Liang Huang, Ge Gao, Chaobo Yang, Xiao-Yan Li, Rui Kai Miao. ✉e-mail: gaetano.magnotti@kaust.edu.sa; sinton@mie.utoronto.ca; ted.sargent@utoronto.ca; xu.lu@kaust.edu.sa

Although these results have shown the impact of pressure on $CO_2R$, the underlying mechanism of the pressure-dependent $CO_2R$ selectivity has yet to be systemically revealed. In particular, the local microenvironment near the $CO_2R$ electrode (such as the concentrations of key species, pHs, etc.) under the influence of pressure is critical to the final $CO_2R$ pathway, but has been rarely observed. This task is beyond the capability of prevailing *operando* tools for electrochemistry, such as Raman spectroscopy[35,36], because the electrode in a high-pressure aqueous-based $CO_2R$ cell is immersed deep inside the liquid electrolyte. The working distance of commercial Raman systems is limited to several millimeters with an excitation power of dozens of milliwatts and the Raman signals of species dissolved in liquids are susceptible to strong background interference[37]. Consequently, reaction mechanisms and cathode design principles relevant to high-pressure $CO_2R$ remain largely unexplored.

Here we examine pressurization in the 1–50 bar range and find that several catalysts, including Cu, Au, Ag, and Sn, become formate selective in aqueous $CO_2R$ systems. Quantitative *operando* Raman spectroscopy, custom-built for high-pressure $CO_2R$ cells, and density functional theory (DFT) calculations, taken together indicate higher $CO_2$ coverage and lower proton concentration on the cathode surface under elevated pressure, each favoring the formate formation. Guided by the pressure-dependent reaction mechanism, we devised a proton-resistant Cu/polypyrrole (Cu/PPy) cathode, which was then assembled into a narrow-gap aqueous flow cell for more selective and active $CO_2$ to formate conversion.

## Results

### Impact of pressure on aqueous-based $CO_2R$

The impact of pressure on $CO_2R$ reaction pathways was evaluated on Cu, Au, Ag, and Sn in a two-compartment high-pressure H-cell (Fig. 1a). We used 0.5 M $KHCO_3$ aqueous solution saturated with $CO_2$ under different pressures as the electrolyte, Pt foil as the counter electrode, and Ag/AgCl (saturated KCl) as the reference electrode. The gas headspace in each compartment was minimal, and three optical windows were fixed in the cathode chamber for ensuing *operando* Raman spectroscopy.

We first prepared a Cu nanoparticle catalyst (Supplementary Fig. 1a). Cyclic voltammetry, x-ray powder diffraction (XRD), and x-ray photoelectron spectroscopy (XPS) indicate metallic Cu(0) as the dominant $CO_2R$ active site (Supplementary Figs. 1b–d)[34]. Transmission electron microscopy (TEM) and high-resolution TEM (HRTEM) determine that the catalyst surface is rich in (111)-oriented planes (Supplementary Figs. 1e, f)[38,39]. In good agreement with the literature[3,40], the as-prepared Cu catalyst converts $CO_2$ to diverse $C_1$ and $C_2$ products under ambient pressure. At −1.1 V versus the reversible hydrogen electrode (vs. RHE), the formate Faradaic efficiency (FE) is only 14.2% and the other $CO_2R$ products (20.5% of CO, 6.6% of $CH_4$, 15.2% of $C_2H_4$, and 8.4% of $C_2H_5OH$) account for 50.7% (Fig. 1b). Surprisingly, at the same cathode potential, the Cu catalyst becomes more formate selective when the pressure increases. Under 50 bar, we note a formate FE of 68.1% and a ten-fold increase in the FE ratio of formate to other $CO_2R$ products, whereas the collective FE of the rest $C_1/C_2$ products drops below 7% (Fig. 1b and Supplementary Fig. 2). The partial current

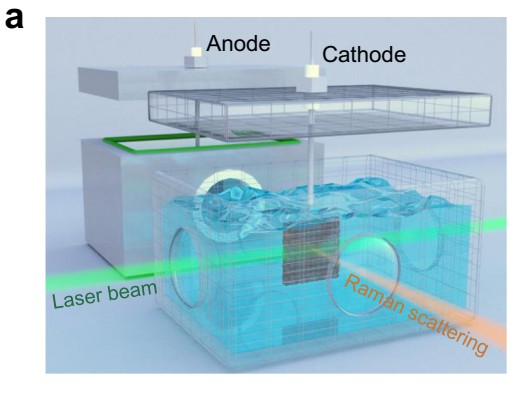

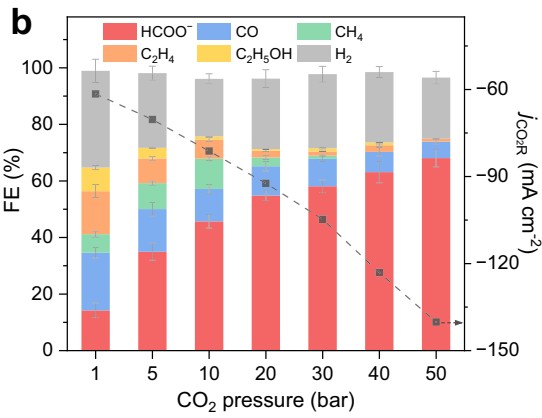

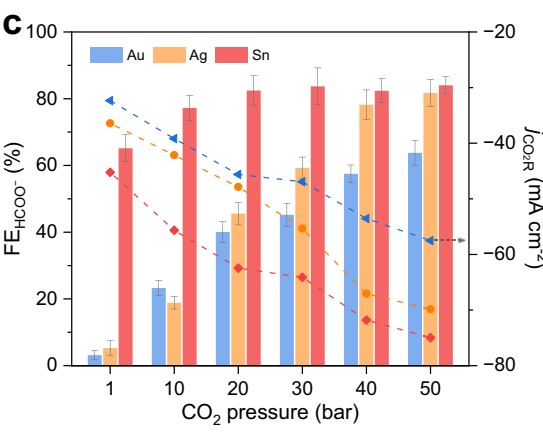

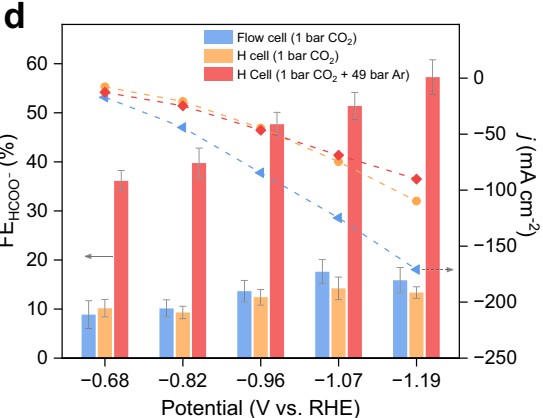

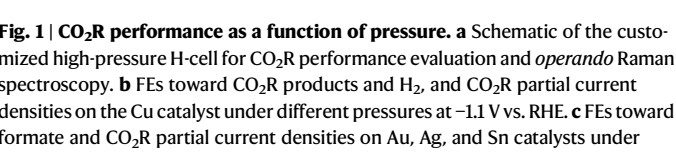

**Fig. 1 | $CO_2R$ performance as a function of pressure. a** Schematic of the customized high-pressure H-cell for $CO_2R$ performance evaluation and *operando* Raman spectroscopy. **b** FEs toward $CO_2R$ products and $H_2$, and $CO_2$ partial current densities on the Cu catalyst under different pressures at −1.1 V vs. RHE. **c** FEs toward formate and $CO_2R$ partial current densities on Au, Ag, and Sn catalysts under different pressures at −1.1 V vs. RHE. **d** FEs toward formate and total current densities on the Cu catalyst in an ambient pressure H-cell (1 bar $CO_2$), an ambient pressure gas-fed flow cell (1 bar $CO_2$), and an H-cell filled with 50 bar $CO_2$/Ar mixture (1 bar $CO_2$ and 49 bar Ar). *j* denotes the current density. Error bars represent the standard deviation of three independent measurements.

density to $CO_2R$ increases from 61.5 mA cm$^{-2}$ under ambient pressure to 140.2 mA cm$^{-2}$ under 50 bar (Fig. 1b).

Similar observations hold for other common $CO_2R$ catalysts targeting CO or formate. We screened Au, Ag, and Sn nanoparticle catalysts (Supplementary Fig. 3) that have been extensively reported[41,42]. At −1.1 V vs. RHE, the Au and Ag catalysts convert $CO_2$ to CO (FE > 62%) with modest yields toward formate (FE < 5%) under ambient pressure (Fig. 1c and Supplementary Fig. 4). Strikingly, at the same cathode potential, both catalysts exhibit a notable increase in formate selectivity under elevated pressure. Under 50 bar, a formate selectivity of 63.8% and 81.7% is achieved on Au and Ag, respectively (Fig. 1c and Supplementary Fig. 4). Likewise, the formate-producing Sn catalyst shows higher formate selectivity under higher pressure (Fig. 1c and Supplementary Fig. 4). The universal increase in formate FEs on Cu, Au, Ag, and Sn suggests that high pressure regulates $CO_2R$ toward the carbon hydrogenation route (*$CO_2$ → *HCOO → HCOO$^-$) regardless of the nature of the catalyst, while compromising the original oxygen hydrogenation pathway (*$CO_2$ → *COOH → *CO/CO) on Cu, Au, and Ag[43]. The trend of the $CO_2R$ partial current density to increase with higher pressure is maintained across all tested catalysts (Figs. 1b, c), implying a denser population of adsorbed $CO_2$ that is accessible to the active sites.

## Mechanistic study

These recurring phenomena motivated us to probe the $CO_2R$ reaction mechanism under the influence of high pressure. We first ruled out the contribution of changes in the catalyst intrinsic properties: Scanning electron microscopy (SEM) and XRD reveal negligible differences in morphologies and crystal structures of the Cu, Au, and Ag catalysts before and after tests under pressure (Supplementary Figs. 5 and 6). After extended $CO_2$ electrolysis under 50 bar, the Cu, Au, and Ag catalysts all resume their original $CO_2R$ behaviors under ambient pressure (Supplementary Fig. 7).

We then assessed the influence of $CO_2$ availability. In a gas $CO_2$-fed flow cell under ambient pressure (Supplementary Fig. 8a), Cu, Au, and Ag manifest regular $CO_2R$ performance with higher current densities compared to ambient pressure H-cell measurements (Fig. 1d and Supplementary Fig. 8b). This implies that the availability of gas-phase $CO_2$ primarily affects $CO_2R$ reactivity rather than the reaction pathway. In another experiment, we saturated the electrolyte in the high-pressure H-cell with a mixture of 1 bar $CO_2$ and 49 bar Ar (Supplementary Fig. 8c), so as to control the $CO_2$ solubility the same as that under ambient pressure. The Cu, Au, and Ag catalysts are more formate selective than the ambient pressure scenarios, albeit to a reduced extent compared to the 50 bar $CO_2$ case (Fig. 1d and Supplementary Fig. 8d). This indicates that, while the pressure-dependent $CO_2R$ performance can be partially explained by the higher availability of dissolved $CO_2$, there may be other critical contributors. We speculate that the pressure might pose impact on the cathode/electrolyte interface and alter the adsorption energy of the intermediates.

To test this hypothesis, we sought to ascertain the species distributions and pH variations in the vicinity of the cathode surface using *operando* Raman spectroscopy. We built a Raman system employing an 18 W continuous-wave laser as the excitation source to realize a >300 mm working distance (Supplementary Fig. 9)[44]. This setup enhanced the signal-to-noise ratio compared to conventional ones: the incident laser beam was highly focused and transmitted along the cathode surface, and the scattered Raman signals were collected, collimated, and screened in the perpendicular direction (Fig. 1a and Supplementary Fig. 9; details in the Supplementary Materials).

With this *operando* Raman platform, we examined the Cu catalyst in 0.5 M KHCO$_3$ saturated with 50 bar $CO_2$[45]. When the laser beam is positioned at the cathode surface (details in the Supplementary Materials), the acquired blended Raman spectrum at −0.9 V vs. RHE displays features of HCOO$^-$, dissolved $CO_2$, and HCO$_3^-$ (Fig. 2a and

Supplementary Fig. 10). In particular, a strong characteristic peak of HCOO$^-$ emerges at 1356 cm$^{-1}$, arising from the C−O symmetric stretch[46], which cannot be observed under the same conditions without applying a potential (Fig. 2a). This HCOO$^-$ peak is rarely reported using commercial Raman spectrometers because it is overlaid by a HCO$_3^-$ peak at 1368 cm$^{-1}$ (Supplementary Fig. 10). When switching to more negative potentials of −1.0 and −1.1 V vs. RHE, the peak intensity of HCOO$^-$ at 1356 cm$^{-1}$ increases whereas that of the dissolved $CO_2$ at 1280 cm$^{-1}$ decreases (insets of Fig. 2a). The HCO$_3^-$ peak intensity at 1019 cm$^{-1}$ varies negligibly throughout the experiment, as indicated by the unchanged peak profile (Fig. 2a). No CO$_3^{2-}$ signal is found, primarily due to the saturation of $CO_2$ (Fig. 2a and Supplementary Fig. 10). No other $CO_2R$ liquid products are detected, such as methanol and ethanol[47], in agreement with the $CO_2R$ performance of Cu under high pressure (Fig. 1b). These results confirm that the carbon source of HCOO$^-$ originates from dissolved $CO_2$, instead of HCO$_3^-$ or CO$_3^{2-}$ in the electrolyte.

We then applied the *operando* Raman system to quantify the HCOO$^-$, dissolved $CO_2$, and HCO$_3^-$ concentrations in the vicinity of the cathode by a hybrid fitting and calibration method (Fig. 2b and Supplementary Figs. 11–13; details in the Supplementary Materials). This allows us to map the key species concentrations as a function of distance ($x$) from the cathode surface ranging from 0 μm (the surface) to 250 μm. At −1.1 V vs. RHE under 50 bar, the HCOO$^-$ concentration decreases from 0.032 M at $x = 0$ μm to 0.016 M at $x = 150$ μm, whereas the dissolved $CO_2$ concentration increases from 0.59 M to 0.72 M (Fig. 2c). The absolute value of the HCOO$^-$ concentration gradient (0.12 mM μm$^{-1}$) in the region of $0 \le x \le 150$ μm is lower than that of the dissolved $CO_2$ (0.81 mM μm$^{-1}$), possibly because the dissolved $CO_2$ is adsorbed on the cathode as a reactant, whereas HCOO$^-$ is released to the electrolyte as a product. We also used the species concentration profiles and equilibrium constants to depict the pH variations as a function of $x$[11,19], suggesting a local pH of 12.3 on the cathode surface.

These findings motivated us to investigate the $CO_2R$ reaction mechanism at high pressure using DFT. In $CO_2R$, oxygen or carbon atoms of $CO_2$ can be protonated to *COOH or *HCOO, respectively— the key intermediates for $CO_2R$ in branching to *CO vs. HCOOH pathways (Supplementary Fig. 14)[48]. In light of the pressure-dependent $CO_2$ solubility and local microenvironment near the cathode surface (Fig. 2c), we sought to explore the impact of $CO_2$ coverage on $CO_2R$ on a Cu(111) facet—the dominant facet of the as-prepared Cu catalyst (Supplementary Fig. 1f). As depicted in Fig. 3a, b, the calculated energy diagrams on the periodic Cu(111) surface indicate the potential-determining steps (PDS) for two $CO_2R$ pathways—the formation of *COOH for *CO pathway (PDS$_{CO}$) and the hydrogenation of *HCOO for HCOOH pathway (PDS$_{HCOOH}$), as seen in previous reports[49]. The free energies of forming *COOH and *HCOO both increase with the $CO_2$ coverage varying from 1/9 monolayer (ML) to 3/9 ML. However, with higher $CO_2$ coverages, the free energy change (ΔG) of PDS$_{CO}$ increases, while that of PDS$_{HCOOH}$ decreases. Specifically, at a $CO_2$ coverage of 1/9 ML, the ΔG of PDS$_{CO}$ is notably lower than that of PDS$_{HCOOH}$, indicating that the *CO pathway is dominant. The situation is reversed when the $CO_2$ coverage gradually increases to 3/9 ML, where the ΔG of PDS$_{CO}$ increases to 1.04 eV and that of PDS$_{HCOOH}$ decreases to 0.75 eV— that means, the *CO pathway become more difficult whereas HCOOH production become more energetically favorable (Fig. 3c). The DFT models reveal that the pressure-dependent $CO_2$ coverage plays a crucial role in shifting the $CO_2R$ product selectivity towards formate/formic acid. It is important to note that due to the imprecise portrayal of carbon-oxygen double bonds in DFT, we focus on the variation trends of the free energies, instead of their absolute values[50]. We then studied the effect of $CO_2$ coverage on the side reaction—the hydrogen evolution reaction (HER; Supplementary Fig. 15). The free energy diagram was calculated based on the Langmuir-Hinshelwood (LH)-type mechanism, and the adsorption of *H was identified as the PDS for HER

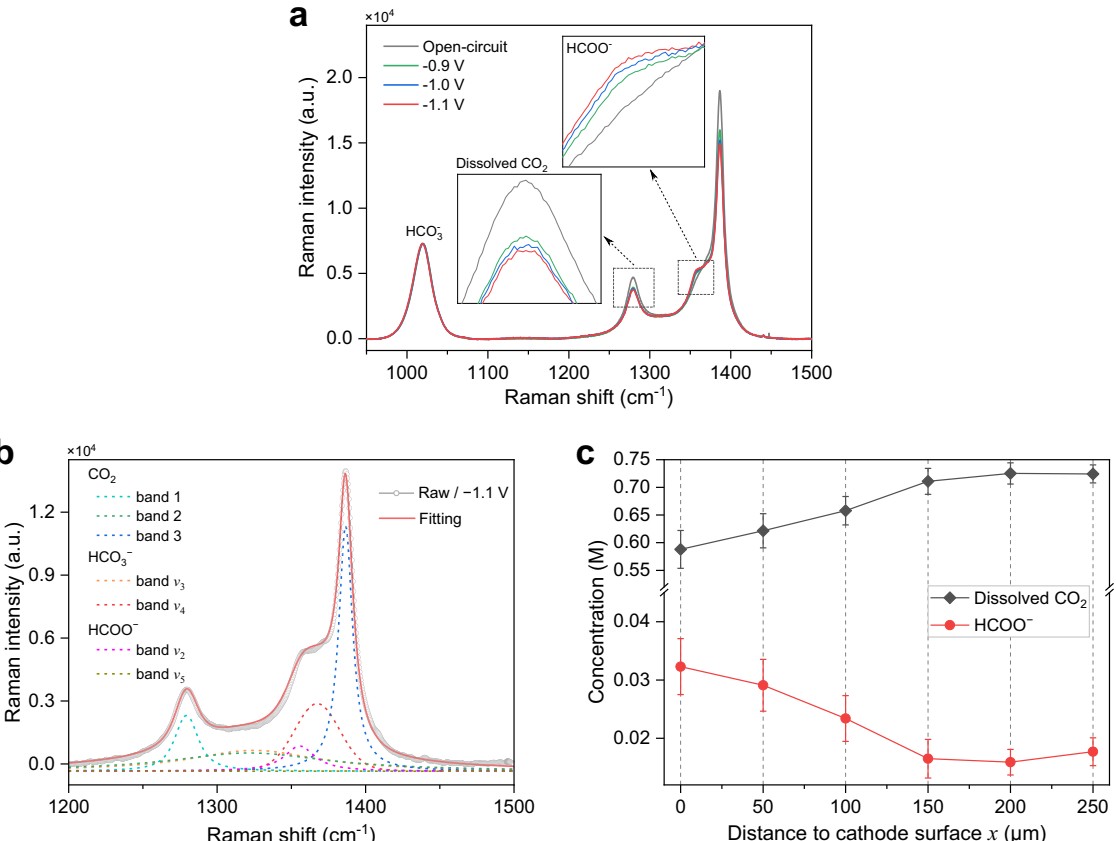

**Fig. 2 | *Operando* Raman spectroscopy for CO₂R at elevated pressure.**
**a** *Operando* Raman spectra acquired on the Cu surface at open-circuit potential, −0.9, −1.0, and −1.1 V vs. RHE under 50 bar. Insets show the characteristic peaks of dissolved CO₂ at 1280 cm⁻¹ (C = O symmetric stretch) and HCOO⁻ at 1356 cm⁻¹ (C − O symmetric stretch). **b** *Operando* Raman spectrum acquired on the Cu

surface at −1.1 V vs. RHE under 50 bar and its fitting curve. Dashed lines mark band assignments for dissolved CO₂, HCO₃⁻, and HCOO⁻. **c** Measured concentrations of dissolved CO₂ and HCOO⁻ against the distance from the Cu surface at −1.1 V vs. RHE under 50 bar. Error bars represent the standard deviation of thirty consecutive measurements.

(PDS$_{H_2}$). As the CO₂ coverage increases from 0 to 2/9 ML, the adsorption strength of *H is slightly decreased, but no further decrease is observed at a higher CO₂ coverage of 3/9 ML (Fig. 3d). The insignificant ΔG of PDS$_{H_2}$ suggests the weak impact of CO₂ coverages on HER, which is consistent with the experimental observation of the slight decrease in H₂ selectivity under elevated pressures. (Supplementary Fig. 16). The interplay of electrochemical measurements, quantitative *operando* Raman studies, and theoretical calculations elucidate how pressure regulates CO₂R pathways: Elevating CO₂ pressure increases the availability of dissolved CO₂, favoring the formate formation and stimulating the CO₂R reactivity. Higher CO₂R reactivity, with its concomitant faster OH⁻ production, alkalizes the microenvironment near the cathode surface.

**Theory-guided electrode design**

The reaction mechanism and experimental results imply that further elevating pressure beyond 50 bar can increase CO₂ coverage and formate productivity (Supplementary Fig. 16). On the other hand, the selectivity of formate under high pressure is predominantly constrained by competing HER, which is weakly influenced by CO₂ coverage and pressure (Fig. 3d). Therefore, we sought to retrofit the Cu cathode surface to suppress HER, so that the formate yield under high pressure can be further improved. We turned our attention to polypyrrole (PPy), capable of limiting the diffusion of excess protons to the electrode surface with its electropositive pyrrole-N group[51,52]. We posited that the controlled assembly of PPy and Cu can lower the local proton concentration near the Cu surface, thus inhibiting the HER (Fig. 4a)[19,53] and promoting the formate selectivity. To test this

postulation, we used an electrochemical anodization method to grow an ultrathin PPy layer on the surface of the Cu catalyst (Supplementary Fig. 17a; details in the Materials and Methods). Fourier transform infrared (FTIR) spectroscopy confirms the formation of PPy (Supplementary Fig. 17b). TEM and scanning TEM (STEM) images, and the corresponding electron energy loss spectroscopy (EELS) mapping indicate an epitaxial growth of PPy on the Cu surface with a thickness <2 nm (Supplementary Fig. 17c, d). The deconvolved N 1*s* and C 1*s* peaks in XPS spectra reveal the presence of polaron (C−N⁺) and bipolaron (C = N⁺) in PPy (Fig. 4b). STEM and HRTEM images show no noticeable structural change of Cu on the as-prepared Cu/PPy catalyst (Supplementary Figs. 18a, b). Cu 2*p* XPS spectra indicate that the PPy layer does not substantially alter electronic structure of the Cu surface (Supplementary Fig. 18c), consistent with the theoretical simulation of the charge density difference (Supplementary Fig. 18d).

Using *operando* Raman spectroscopy, we determine that the local HCOO⁻ concentration on the Cu/PPy surface (0.057 M) is higher than that of Cu (0.032 M) at −1.1 V vs. RHE under 50 bar, validating the function of PPy to promote formate production (Supplementary Fig. 19). This phenomenon translates to other cathode potentials ranging from −0.9 to −1.2 V vs. RHE (Supplementary Fig. 19). We then evaluated the CO₂R performance of Cu/PPy in the high-pressure H-cell. The Cu/PPy catalyst exhibits a notable increase in formate selectivity and productivity compared to Cu over a wide range of pressures from 10 to 50 bar (Fig. 4c) at −1.1 V vs. RHE. Under 50 bar, the FE toward formate surpasses 82% at −1.1 V vs. RHE, and the formate partial current density exceeds 200 mA cm⁻² at −1.21 V vs. RHE (Fig. 4d). In contrast, the Cu catalyst is limited to FEs <70% and partial current densities

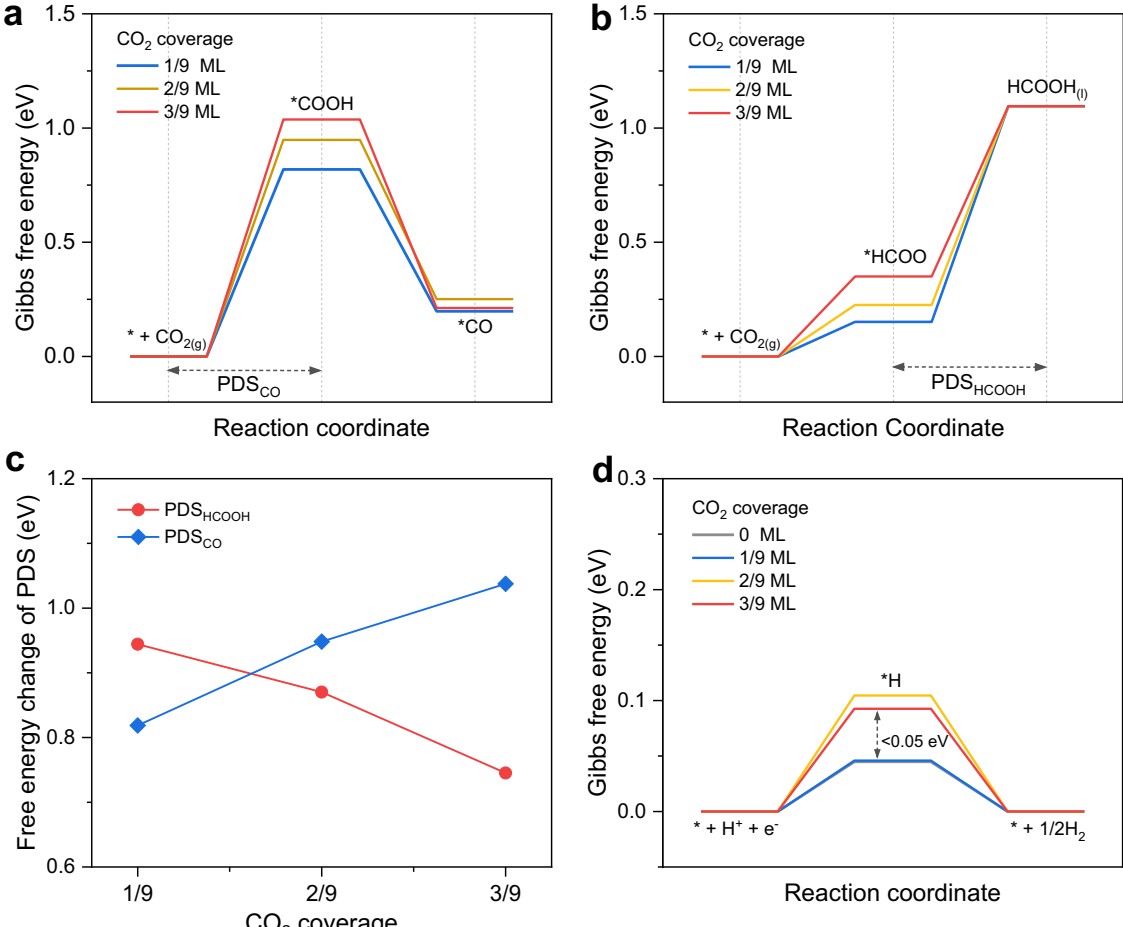

**Fig. 3 | DFT simulations of the pressure-mediated CO₂R mechanism. a** Free energy diagram from $CO_2$ toward *CO. **b** Free energy diagram from $CO_2$ toward HCOOH. **c** Free energy changes of PDS in $CO_2$R for *CO or HCOOH production on Cu(111) with various $CO_2$ coverages. **d** Free energy diagram of HER on Cu(111) with various $CO_2$ coverages.

<130 mA cm⁻² toward formate (Fig. 1b). Bare PPy, on the other hand, shows almost no $CO_2$R activity on its own (Supplementary Fig. 20), something we attribute to its weak adsorption of $CO_2$R intermediates (Supplementary Fig. 18e).

We then integrated the Cu/PPy cathode into a narrow-gap aqueous flow cell (Fig. 5a, Supplementary Fig. 21a, b). The flow cell employed 1 M $KHCO_3$ and 0.5 M $K_2SO_4$ saturated with 50 bar $CO_2$ as the catholyte and anolyte, respectively, and $RuO_2$/Ti foam as the anode. The catholyte and anolyte channels were ultraslim (~0.3 mm) to minimize the ohmic loss (Supplementary Fig. 21c). The narrow-gap aqueous flow cell manifests a maximal formate FE of 84.7% with a full cell voltage of 2.85 V at 200 mA cm⁻² (Fig. 5b and Supplementary Fig. 22a). The formate partial current density reaches 310 mA cm⁻² with a FE of 77.5% and a cell voltage of 3.85 V at 400 mA cm⁻² (Fig. 5b and Supplementary Fig. 22a). The cell voltage and formate FE remain largely stable over the course of a 12 h chronopotentiometric operation at 400 mA cm⁻² (Fig. 5c). Formate is confirmed to be the only liquid product (Fig. 5b and Supplementary Fig. 22b).

## Discussion

In summary, we report here on the role of pressure in regulating aqueous-based $CO_2$R pathways that is catalyst independent. We developed *operando* methods for high-pressure conditions, including a quantitative *operando* Raman system capable of probing the local microenvironment near the electrode in high-pressure aqueous-

based $CO_2$R cells. The work sheds light on cathode design principles and suggests further avenues for commodity chemicals from $PCO_2$.

## Methods

### Electrodes preparation

Metal electrodes were prepared using galvanostatic electrodeposition in a three-electrode setup using an electrochemical workstation (Bio-Logic SP-150 Potentiostat). Commercial Cu foam (MTI Corporation, 99.9%, 0.30 mm thick) or carbon paper (Toray 120, Fuel Cell Store, 0.30 mm thick) was used as the working electrode. An Ag/AgCl electrode (CH Instruments, saturated KCl) was used as the reference electrode, and a Pt foil (Tianjin Aida Hengsheng Technology Development Co., Ltd, >99.99%, 1 × 1 cm) was used as the counter electrode. All chemicals and reagents were used as received without further purification.

For the Cu electrode, a piece of Cu foam (1 × 2 cm) was washed by acetone (Sigma-Aldrich, ≥ 99.5%), ethanol (Sigma-Aldrich, ≥99.8%), 0.5 M HCl (Sigma-Aldrich, 37%), and deionized water (Millipore, 18.2 MΩ cm) successively, each for 5 min under sonication. The Cu foam was then electrochemically anodized in 1.0 M KOH (Sigma-Aldrich, ACS reagent) for 20 min at 10 mA cm⁻² to obtain $Cu(OH)_2$ nanowire arrays. Finally, the sample was electrochemically reduced in 0.5 M $KHCO_3$ (Macklin Inc., ≥99.9%) for 20 min at −5 mA cm⁻² to form the Cu electrode. For Cu/PPy electrode, the aforementioned $Cu(OH)_2$ nanowire arrays were further anodized in 0.1 M pyrrole (Alfa Aesar, > 98.0%) and 0.01 M KOH for 2 min at 5.0 mA cm⁻², followed by being

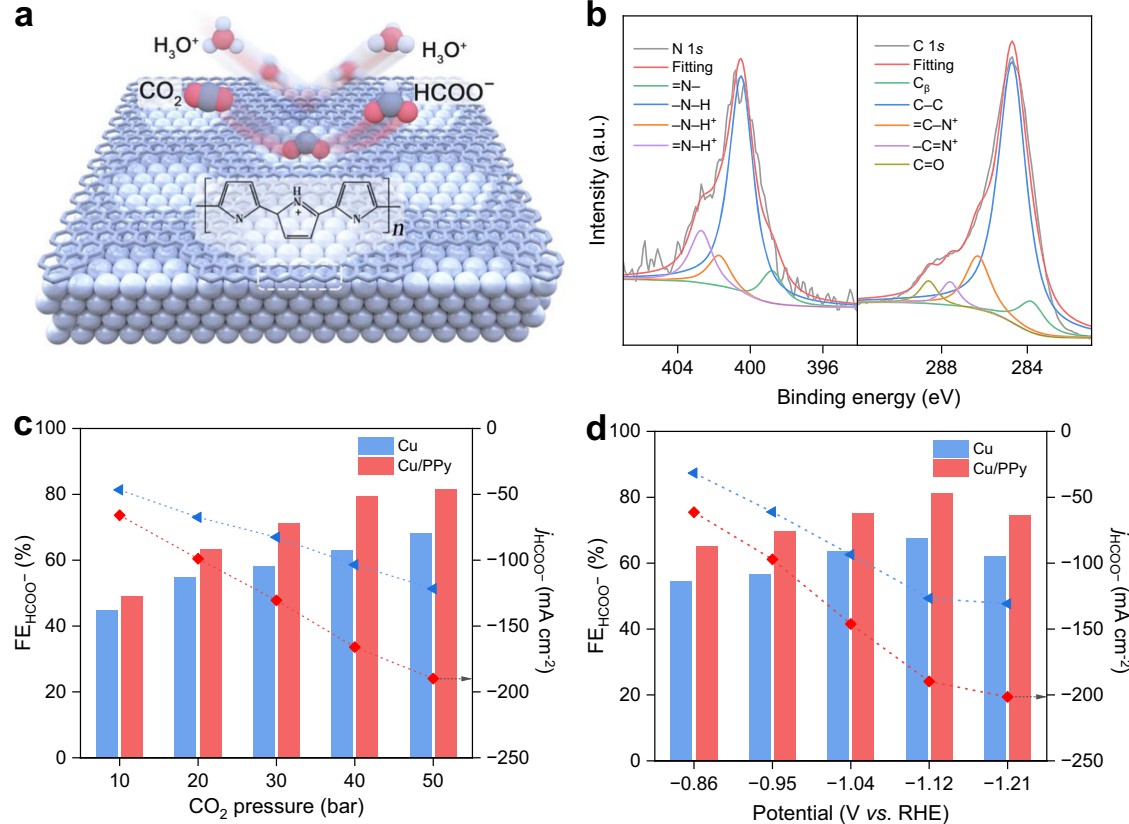

**Fig. 4 | Enhanced formate production on Cu under high pressure enabled by a proton-resistant layer. a** Schematic of the proton-resistant cathode surface functionalized by the PPy layer. **b** N 1$s$ and C 1$s$ XPS spectra of the Cu/PPy catalyst.

FEs and partial current densities toward formate on Cu and Cu/PPy catalysts as a function of **c** pressure at −1.1 V vs. RHE, and **d** cathode potential under 50 bar. $j$ denotes the current density.

reduced in 0.5 M KHCO₃ for 20 min at −5 mA cm⁻² (Supplementary Fig. 17a).

For the Au electrode, the dendritic Au nanoparticles (Supplementary Fig. 3a) were formed on carbon paper (1 × 2 cm) through galvanostatic electrodeposition for 20 min at −5.0 mA cm⁻². The electrolyte solution consisted of (i) 50 mM HAuCl₄ (Alfa Aesar, 99.9%) dissolved in 0.1 M H₂SO₄ (Sigma-Aldrich, 95.0–98.0%) as the Au precursor and (ii) 0.15 mM Pluronic F-127 (Innochem Co., Ltd., average molecular weight ~10,000) as the structure-directing agent.

For the Ag electrode, the dendritic Ag nanoparticles (Supplementary Fig. 3b) were formed on carbon paper using the same protocol as the Au electrode, except for the electrolyte solution. The electrolyte solution was 50 mM AgNO₃ (Sigma-Aldrich, ≥99.0%) dissolved in 0.4 M NH₃·H₂O (Sigma-Aldrich, 28.0–30.0% NH₃ basis) and 0.15 mM Pluronic F-127.

For Sn electrode, the hierarchical flake-like Sn nanoparticles (Supplementary Fig. 3c) were formed on carbon paper using the same protocol as the Au and Ag electrodes, except for the electrolyte solution. The electrolyte solution was 50 mM SnCl₂ (Innochem Co., Ltd., 99%) dissolved in 0.4 M Na₄P₂O₄ (Sigma-Aldrich, ≥95.0%) and 0.15 mM Pluronic F-127.

## Material *characterization*

Scanning electron microscopy (SEM) images were collected using FEI Quanta 600 FEG ESEM operated at 15 kV. Transmission electron microscopy (TEM) images were acquired using FEI Tecnai G² Spirit Twin operated at 120 kV. High-resolution TEM (HRTEM), scanning TEM (STEM), and electron energy loss spectroscopy (EELS) were performed by FEI Titan 80-300 equipped with a field emission gun and spherical aberration corrector operated at 300 kV. EELS mapping was collected

using a post-column filter in diffraction mode. X-ray photoelectron spectroscopy (XPS) was operated using Kratos Analytical AMICUS/ ESCA 3400 equipped with an Mg-anode Kα excitation x-ray source ($hv$ = 1253.6 eV) at 10 kV, 10 mA, and 2×10⁻⁶ Pa. The measured binding energies were calibrated based on C 1$s$ binding energy at 284.8 eV. X-ray powder diffraction (XRD) was carried out using the Bruker D8 Advance with a Cu Kα radiation. Fourier transform infrared (FTIR) spectroscopy was performed using the Thermo Scientific Nicolet 6700 FTIR spectrometer.

## High-pressure H-cell

The customized two-compartment high-pressure H-cell (Fig. 1a and Supplementary Fig. 8c) was made of Teflon-lined titanium. The two compartments were separated by a proton exchange membrane (Nafion 117, Fuel Cell Store). Each compartment contained 120 mL of 0.5 M KHCO₃ aqueous solution with a gas headspace <15 mL, and was connected to an independent pressure regulator. The working electrode (1 × 0.5 cm) and reference electrode (Ag/AgCl with saturated KCl, Gaoss Union) were placed inside the cathode compartment, and the counter electrode (Pt foil, 1 × 2 cm) was placed inside the anode compartment. A small hole (diameter <0.5 mm) was drilled on the top of the reference electrode to balance its internal and external pressure. Prior to the tests, electrolytes in each compartment were first purged for 5 min using the feed gas (99.995% CO₂ or its mixture with 99.999% Ar, Air Liquide), and then saturated with the feed gas by stirring for 30 min to reach equilibrium under the desired pressure (from 1 to 50 bar). The pressure of the two compartments were kept identical. The gas products in the headspace of the cathode compartment were sampled using 10 mL air-tight syringes from an outlet relief valve, and then injected into a gas chromatograph (GC) system.

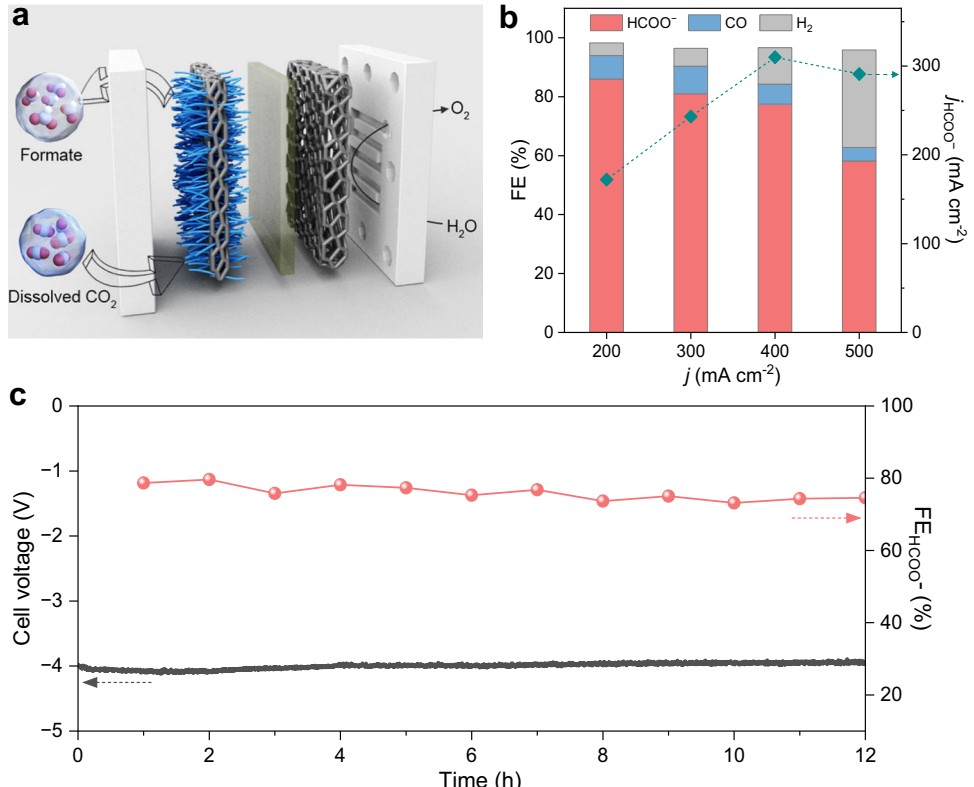

**Fig. 5 | Electrolysis of PCO₂ in narrow-gap aqueous flow cell. a** Schematic of the high-pressure narrow-gap aqueous flow cell. **b** FEs and partial current densities at total current densities of 200, 300, 400 and 500 mA cm⁻² on Cu/PPy in the high-pressure narrow-gap aqueous flow cell fed by 50 bar PCO₂. No other liquid products were detected. *j* denotes the current density. **c** Prolonged electrolysis of 50 bar PCO₂ to formate on Cu/PPy in the narrow-gap aqueous flow cell at a constant current density of 400 mA cm⁻². The full cell voltage and formate FE were well retained over 12 h continuous operation.

## Ambient-pressure gas-fed flow cell

The customized ambient-pressure gas-fed flow cell comprised three compartments made of poly(methyl methacrylate) (PMMA): a gas compartment with serpentine channel, a catholyte compartment, and an anolyte compartment (Supplementary Fig. 8a). The size of each compartment was 0.5 cm (height) × 0.5 cm (width) × 1 cm (length). The working electrode was housed between the gas and catholyte compartments, with the catalyst layer facing the catholyte. An Ag/AgCl electrode (saturated KCl) and a RuO₂/Ti foam (Gaoss Union, 1.5 × 1.5 cm) were used as the reference and counter electrode, respectively. The anolyte and catholyte compartments were separated by a Nafion 117 membrane. 0.5 M KHCO₃ electrolytes were supplied to the catholyte and anolyte compartments and recirculated at a constant flow rate of 10 mL min⁻¹ using a peristaltic pump (Longer Pump, BT100-2J). CO₂ was delivered to the gas compartment at a constant flow rate of 20 sccm using a mass flow controller (Cole-Parmer, Masterflex Proportional Flowmeter Controller), and the gas effluents were extracted for detection by GC.

## High-*pressure* narrow-gap aqueous flow cell

The high-pressure narrow-gas aqueous flow cell system (Supplementary Fig. 21a) consisted primarily of a narrow-gap aqueous flow cell, two high-pressure high-performance liquid chromatography pumps (HPLC pump; Sanotac SP6010), two backpressure valves (Beijing Xiong Chuan Technology Co. LTD), two Teflon-lined titanium tanks containing 1.0 M KHCO₃ catholyte and 0.5 M K₂SO₄ anolyte, respectively, and a CO₂ gas cylinder. The high-pressure narrow-gap aqueous flow cell was assembled by stacking the following components in order: a Cu/PPy cathode sandwiched by two polytetrafluoroethylene (PTFE) gaskets (with a 0.5 × 1 cm window as the reactive area) as the catholyte compartments, a Nafion 117 membrane, and a RuO₂/Ti foam

anode sandwiched by two aforementioned PTFE gaskets as the anolyte compartments (Fig. 5a and Supplementary Fig. 21b). These components were fixed and sealed by two titanium plates with channels and ports. The catholyte and anolyte were pressurized by high-pressure HPLC pumps and equilibrated by backpressure valves. Prior to each experiment, the air in the catholyte was purged out by bubbling CO₂ under atmospheric pressure. Then, the pressures of CO₂ and electrolytes were simultaneously and gradually increased by adjusting the gas cylinder valve and backpressure valves. Both cathode and anode compartments of the narrow-gas aqueous flow cell were pressurized to 50 bar and held for 30 min to achieve the equilibrium solubility of CO₂ in catholyte. During CO₂R, the electrolytes were recirculated at a constant flow rate of 10 mL min⁻¹. The anodic O₂ and cathodic CO₂/ CO₂R products were discharged separately through the outlets of each backpressure valves. The CO₂R gas products were collected by a syringe and analyzed by GC, while the liquid products were sampled by withdrawing the catholyte solution through the sampling port every hour and analyzed by NMR.

## Electrochemical measurements

All electrochemical measurements were conducted using an electrochemical workstation (BioLogic SP-150 Potentiostat) at room temperature. Cathode potentials in three-electrode systems (H-cells and gas-fed flow cell) were recorded with iR compensation, where the cell resistance was determined using a current-interrupt method, and the potential was manually corrected after each measurement. And then converted to the reversible hydrogen electrode (RHE) scale using the following equation:

$$E_{RHE} = E_{Ag/AgCl} + 0.197V + 0.059 \times pH \qquad (1)$$

Full-cell voltages of the high-pressure narrow-gap aqueous flow cell were measured using the chronopotentiometry method. The currents were normalized to the geometric area of the working electrodes.

## Product analysis

Gas products were analyzed using a GC (Trace 1310, Thermo Scientific) equipped with Molecular Sieve 5 A and Porapak N columns. Ar (Al Khafrah Industrial Gases, 99.999%) was used as the carrier gas. CO, $CH_4$, $C_2H_4$, $C_2H_6$, $C_3H_6$, and $C_3H_8$ were quantified using a flame ionization detector (FID) with a methanizer. $H_2$ was quantified using a thermal conductivity detector (TCD). The volumes of gas products were derived from the output peak areas based on calibration curves.

Liquid products were analyzed by a $^1H$ NMR (Bruker, 600 MHz) using water suppression method. Each liquid sample was prepared by mixing 490 μL of the electrolytes with 110 μL of the internal standards (20 ppm of dimethyl sulfoxide in $D_2O$).

The Faradaic efficiency (FE) of a specific product (p) was calculated using the following equation:

$$FE_p(\%) = \frac{z \times n \times F}{Q} \times 100\% \qquad (2)$$

where z denotes the number of the electrons transferred to one p molecule, n represents the total moles of the product, F is the Faradaic constant (F = 96,485 C mol$^{-1}$), and Q indicates the total number of electrons transferred.

## Operando Raman spectroscopy

The schematic of the custom-built *operando* Raman system is illustrated in Supplementary Fig. 9. The high-pressure H-cell was placed on a mechanical sample stage. The excitation source was a 532 nm continuous-wave narrow-band laser (Coherent, Inc. Verdi G18) with a maximum power of 18 W. The laser was highly focused by a spherical convex lens (focus length f = 500 mm) in the probing region and transmitted along the cathode surface. To locate the cathode surface (x = 0 μm), we first moved the sample stage until the laser beam was cut by the cathode, and then moved the sample stage backward until the laser beam fully appeared. The species concentrations against the distance from the cathode surface were acquired by controlling the sample stage for line scan. The Raman signal was collected by a Nikon micro lens (f = 105 mm, F#2.8) through the perpendicular window of the cathode compartment (Fig. 1a, Supplementary Figs. 8c and 9) and collimated by a digital single-lens reflex (DSLR) prime lens (Samyang, f = 135 mm, F#2.2). The combination of a 532 nm notch filter with a half-wave plate (HWP) and a wire grid polarizer was placed in the collimated beam to filter out the Rayleigh signal and stray light background. The beam was then rotated by 90° and passed through the slit of an astigmatism-free spectrometer (Princeton Instrument Isoplane 320). The design of the spectrometer avoided the bowing effect and allowed the integration along the spatial direction without degrading the spectral resolution. An electron multiplication charge-coupled devices (EMCCD) camera (Princeton Instruments, ProEM:1600 200) imaged the Raman signal with a 0.01 nm per pixel dispersion along the spectral direction. Full binning along the laser propagation direction and 1 s exposure time were applied to ensure a high signal-to-noise ratio[54]. The spatial resolution along the laser propagation was about 8.6 mm, and the resolution along the vertical direction was around 50 μm.

As depicted in Supplementary Fig. 10, some Raman peaks of the key species ($HCO_3^-$, $HCOO^-$, and dissolved $CO_2$) overlapped. We therefore employed a hybrid fitting and calibration method to rigorously convert the areas under the Raman peak to concentrations.

First, the Raman spectra were acquired for standard aqueous solutions of $KHCO_3$, HCOOK, and dissolved $CO_2$ with known concentrations (Supplementary Fig. 11). Taking HCOOK as an example, the spectra of its $v_2$ and $v_5$ bands at a concentration of 0.4 M were fitted with the summation of the corresponding Voigt functions (Supplementary Fig. 12a), which settled the parameters of the center wavelength, Raman line shape, and area ratio of the two bands. Then, we fitted the spectra of 0.025, 0.05, 0.1, 0.2 and 0.3 M HCOOK based on the as-obtained parameters and set the peak area as the free parameter. In this way, the relationship between the area of fitted function and the $HCOO^-$ concentration was established (Supplementary Fig. 13a). Following the same procedure, we can quantify the concentrations of other species in the vicinity of the cathode surface.

## Theoretical methods

All density functional theory (DFT) calculations were performed by the Vienna ab initio simulation program (VASP)[55,56]. The core-valence interactions were calculated by the project augmented wave (PAW) method with 450 eV as the cut-off energy[57,58]. The generalized gradient approximation in the Perdew–Burke–Ernzerhof functional (GGA-PBE) was applied to describe the exchange-correlation correction effect[59]. The DFT-D3 method was used to consider the dispersion correction of the van der Waals force[60]. For the geometry optimization, the self-consistent iteration must reach $10^{-6}$ eV for the energy convergence and 0.01 eV Å$^{-1}$ for the force convergence.

The (3 × 3) Cu(111) model consisted of four Cu atomic layers, where the two bottom layers were fixed to mimic the bulk material and the rest of atoms were relaxed. To account for both explicit solvation and field effects, we incorporated one charged water layer onto the Cu(111) surface at the intermediates according to studies of Nørskov et al.[61], where the optimal water structure are obtained via using a minima-hopping algorithm[49]. Here, the periodic structure of one charged water layer, consisting of five water molecules and one hydronium molecule in the (3 × 3) cell, closely resembles the hexagonal ice-like structure using previously in various DFT-based studies of adsorption and proton-coupled electron transfer (PCET) kinetics on Pt(111)[62,63], and has been widely used in the $CO_2R$ studies[64]. The vacuum space was set at about 15 Å in z-axis to avoid interactions between the periodic images. To investigate the impact of $CO_2$ coverage, we conducted a comprehensive analysis involving the adsorption of one, two, or three $CO_2$ molecules onto the periodic (3 × 3) Cu(111) surface. Two key reaction pathways of $CO_2R$ toward *CO or HCOOH were considered as follows:

$CO_2$ to HCOOH:

$$* + CO_{2(g)} + H^+ + e^- \leftrightarrow *HCOO \qquad (3)$$

$$*HCOO + H^+ + e^- \rightarrow HCOOH_{(l)} \qquad (4)$$

$CO_2$ to *CO:

$$* + CO_{2(g)} + H^+ + e^- \leftrightarrow *COOH \qquad (5)$$

$$*COOH + H^+ + e^- \rightarrow *CO + H_2O_{(l)} \qquad (6)$$

Additionally, the elementary steps of HER were listed as below:

$$* + H^+ + e^- \leftrightarrow *H \qquad (7)$$

$$*H + H^+ + e^- \rightarrow H_{2(g)} \qquad (8)$$

where * represents the active site or the adsorbed intermediate. The proton-electron pair was treated with the computational hydrogen electrode (CHE) model[49]. The PDS in the three reaction pathways was identified based on the most positive change in free energy. A more positive change in PDS indicated a more difficult reaction pathway. The optimized structures for all reactions were presented in the Supplementary Figs. 14 and 15. The free energy change of each elementary step for the production of *CO, HCOOH$_{(l)}$, or H$_{2(g)}$ was obtained by the correction of Gibbs free energy ($G$) at room temperature (T = 298.15 K), using the following equation:

$$G = E_{DFT} + ZPE + \int C_p dT - TS \qquad (9)$$

where $E_{DFT}$ is the energy changes of DFT calculations, $ZPE$ is the zero-point energy, $C_p$ is the heat capacity, and $S$ is the entropy.

## Data availability
The data that support the findings of this study are available from the corresponding authors upon reasonable request.

## Code availability
Vienna ab initio simulation package (VASP) for the DFT calculations is available at https://www.vasp.at/.

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

## Acknowledgements

This work was financially supported by the Baseline Fund (BAS/1/1413-01-01) and Center Competitive Fund (URF/1/1975-16-01) to X.L. from King Abdullah University of Science and Technology (KAUST). E.H.S. and D.S. acknowledge the Natural Sciences and Engineering Research Council (NSERC) of Canada and the Ontario Research Fund: Research Excellence Program. DFT calculations were performed on the Niagara supercomputer at the SciNet HPC Consortium. G.M. thanks the KAUST Research Funding Office (URF/1/3715-01-01 and BAS/1/1388-01-01).

## Author contributions

X.L., E.H.S., D.S., and G.M. supervised the project. X.L. and L.H. conceived the idea. L.H. and G.G. fabricated and characterized the samples and conducted electrochemical experiments. C.Y. and L.H. carried out the operando Raman measurements. X.Y.L. and P.O. performed the theoretical calculations. R.K.M. and Y.X. assisted the electrode assembly in the narrow-gap aqueous flow cell. C.T.Y., Y.H., R.K.M., and K.X. contributed to data analysis and manuscript editing. L.H. and X.L. wrote the manuscript. All authors discussed the results and assisted with the manuscript preparation.

## Competing interests
