## [Peer review file · Nature Communications]

REVIEWER COMMENTS

Reviewer #1 (Remarks to the Author):

This manuscript from Liang Huang et al. discusses a very interesting aspect, the high-pressure electrolysis of CO₂. The experiments are interesting, and the used methods are suitable for this study. Despite of this, I still believe that this manuscript is premature. The first half of the manuscript deals with the effect of pressure, while the second half demonstrates the use of a polymer coating to change the selectivity of CO₂R to formate. In opposite to the authors, I do not see a clear connection between these halves. A surface coating can have a multitude of different effects on the catalysis process, of which only the H⁺ repellent property of PPy is highlighted. This is unfounded and misleading.

Based on all above, I believe that the authors should work on both parts to make it a convincing and coherent manuscript. Therefore I suggest the major revision of the manuscript.

My specific concerns:

- Scheme 1. is misleading, as pressurization of the CO₂ stream is indicated, while the use of pressurized sources is depicted in the Introduction. Pressurizing CO₂ is associated with further energy costs.
- A local pH of 12.3 is reported at high pressure, and it is claimed that it is higher compared to the case of atmospheric pressure. Is this really the case, if we apply the same current density? This should be supported by experimental results.
- The formation of OH⁻ is said to be a parasitic process. It is not parasitic, it is part of the stoichiometry of CO₂ reduction.
- The theory of needing a hydronium-repellent layer to increase formate selectivity seems a bit bizarre, as Cu catalysts are typically used in 1M (or more concentrated) alkaline electrolyte solutions, where very little formation production is observed. Considering this, I do not really see the relevance of the DFT studies where hydronium ions are brought to the surface.
- PPy is deposited oxidatively. This can alter the surface of the catalyst. This must be addressed
- PPy might bind CO₂, or some reaction intermediates, that can alter the course of the reaction.
- Coating the catalyst with PPy might alter its electronic structure, that can also change the reaction selectivity.
- Was the current density identical when comparing the surface pH of the bare and PPy coated catalysts?
- HCOO⁻ relative purity is shown in Fig 5. I understand that this is related to the liquid products, but I still find it misleading. The formate concentration should be presented.

Reviewer #2 (Remarks to the Author):

Summary of the paper: The authors investigate how pressurization affects the CO₂RR. To this end, they developed new operando IR experimental techniques. This allows to study the electrochemical reaction under pressure. They find that higher pressure alters the mechanism to favor formate, most likely due to a different ratio of CO₂* and hydronium ions. They employ both computational and experimental techniques to elucidate the different product selectivity and rationally improve the catalyst based on their mechanistic findings.

Significance:

Their significance for the catalysis community is very high due to several reasons:

First, the authors develop new operando IR techniques which can provide crucial mechanistic insights for pressurized electrochemical experiments. This type of method development is crucial for obtaining insights in catalytic mechanisms.

Second, this study is a great example of how the combination of experimental and computational insights can be utilized to rationally improve the catalyst.

Unfortunately, I don't believe that this study is suitable for a broad audience such as Nature Communications. The reasons are outline below:

First, optimizing reaction conditions to yield 2 electron products such as CO or CHOOH has been studied in great detail in the literature. Altering the product distribution to yield four or six electrons such as C₂ products would be highly desirable.

Second, while the product distribution of the CO₂RR changes using pressure the selectivity of CO₂RR versus HER is not strongly affected as seen in Fig. 1b.

Third, the effect that pCO₂RR changes the product distribution has been observed before (as acknowledge by the authors).

Major points:

The authors developed new experimental techniques which give crucial mechanistic insights.

This guided a well designed DFT study. In turn, those insights are indirectly confirmed by rational modification to the catalyst by introducing PPy further improving the performance. I really enjoyed reading this manuscript.

One concern I have regarding the DFT simulation are the ddG plots in Fig 3 a and b. The predicted results do not follow the experimental FE in 1b. The ddG changes only slightly as function of CO₂* coverage and even at 1/9 the difference is larger than 0.5 eV suggesting that formic acid should already be the major product. The author should add clarifying statements about expected accuracies of the used DFT framework and cite similar studies. In addition, it should not be presented as a driving indicator for the change in product distribution.

Another interesting point omitted in their simulations is HER side reaction. If the hydronium ions dictate the product distribution of the CO₂RR, a similar effect should be observed for HER activity. It seems not intuitive to me that this is not the case. Simulation should be carried out to investigate this (perhaps in a separate study).

Minor points:

The choice of 6 water molecules seems a bit arbitrary, the authors should justify this choice and show how their findings converge with respect to the number of water molecules. The same should be done for adding hydronium ions, it is not clear how many were added and several should be tested and reported.

The axis labels and ticks could be larger in the images

Fig 4b does not give a lot of insight and can be moved into the SI

Scholar representation:

Well written, very clear figures and amazing illustrations. The authors should cite more work about the pressurized CO₂RR such as the original report from the 1990s.

Reviewer #3 (Remarks to the Author):

In the article " Pressure dependence in aqueous-based electrochemical CO₂ reduction", the authors have discussed the effect of the pressure on the CO₂ electrochemical process on Cu, Ag, Au and Sn cathode. Electrochemical reduction of CO₂ is among the most successful methods of conversion of this waste gas to high value-added products and the process is under intense investigation by many researchers. The manuscript is well-written and concise. Though the results are very interesting some questions/concerns need to be addressed before the final acceptance.

- - This work suffers from a lack of previous important achievements reported in the literature by several authors on the electrochemical CO₂ conversion under high pressure of CO₂. First of all, the authors report that they "discover that pressurization up to 50 bar regulates common CO₂R catalysts to be formate selective in aqueous systems", however, the effect of the CO₂ pressure on the selectivity of the process as well as on the mechanism reaction under high pressure was already reported. In 1993, Kudo et al. (J. Electrochem. Soc. 140 (1993) 1541, doi:10.1149/1.2221599) showed that an increase of CO₂ pressure allowed to favor CO₂ reduction to formic acid using a Ni wire electrode. Similar results were reported by Hara et al. (J. Electroanal. Chem. 391 (1995) 141–147. doi:10.1016/0022-0728(95)03935-A.) and Mizuno et al. (Energy Sources 17 (1995) 503–508. doi:10.1080/00908312.). As well as interesting results on the reaction mechanism of the CO₂ conversion under high pressure were found for example by Proietto et al. (ChemElectroChem 6 (2019) 162–172, doi:10.1002/celec.201801067) and Morrison et al. J. Electrochem. Soc. 166 (2019) E77–E86. doi:10.1149/2.0121904jes). Other citations in this field are missing; see for examples: Fresenius Environmental Bull 12 (2003) 1202–1206; Appl. Catal. A Gen. 274 (2004) 237–242; Journal of The Electrochemical Society, 159 (9) (2012) F514-F517; Energy Environ. Sci., 11 (2018) 2531; Ind. Eng. Chem. Res. 58 (2019) 1834–1847; Electrochim. Acta 199 (2015) 332–341; Journal of CO₂ Utilization 67 (2023) 102338; and so on. Hence, the authors should consider improving the introduction section and to discuss the results according to the literature.

- Please, consider that reference 13 seems not appropriate to the context in the following sentence: "However, industry works exclusively with pressurized CO₂ (PCO₂) in capture, transport and storage. In many such cases, the CO₂ is in dissolved form (1 - 110 bar)^{13,14}." Hence, I would suggest clarifying this content or removing the citation.

- The authors should explain why they used 1 bar of CO₂ and 49 bar of Ar. Why the choice to investigate the process under a final pressure of 50 bar?

- One of the main technological problems on the electrochemical cell under high pressure is the leakage of gas; the author should give more information of the high-pressure narrow-gap aqueous flow cell. It is not possible to understand how the pressure balance was assured between the two compartments and how you can reproduce these experiments.

Responses to Reviewers

We would like to express our gratitude to the Reviewers for their immense time and efforts in reviewing our manuscript. Their comments have greatly helped us to constructively modify our manuscript. A point-by-point response to all comments is given below, and the corresponding changes are highlighted in the revised manuscript and supplementary materials.

REVIEWER COMMENTS

Reviewer #1 (Remarks to the Author):

This manuscript from Liang Huang et al. discusses a very interesting aspect, the high-pressure electrolysis of CO₂. The experiments are interesting, and the used methods are suitable for this study. Despite of this, I still believe that this manuscript is premature. The first half of the manuscript deals with the effect of pressure, while the second half demonstrates the use of a polymer coating to change the selectivity of CO₂R to formate. In opposite to the authors, I do not see a clear connection between these halves. A surface coating can have a multitude of different effects on the catalysis process, of which only the H⁺ repellent property of PPy is highlighted. This is unfounded and misleading.

Based on all above, I believe that the authors should work on both parts to make it a convincing and coherent manuscript. Therefore I suggest the major revision of the manuscript.

Response: We are very grateful to the reviewer for her/his appreciation of our work. Following the reviewer's insightful and constructive comments, we have conducted additional experiments and theoretical analysis, and systematically revised the manuscript to address the reviewer's concerns.

Meanwhile, we endeavor to reinforce the connection between the two sections in the manuscript with additional analysis and discussions. For instance, in the first section, we added new DFT results to demonstrate that the CO₂ coverage mainly affected CO₂R, whereas it can barely influence HER – that means, the selectivity of formate under high pressure was predominantly limited by HER. Guided by this mechanism, in the second section, we showed that incorporating a proton-resistant PPy layer can reasonably suppress HER and promote the formate selectivity. This finding connects the high pressure approach to the need for the layer developed here.

Specific concerns:

- Scheme 1. is misleading, as pressurization of the CO₂ stream is indicated, while the use of pressurized sources is depicted in the Introduction. Pressurizing CO₂ is associated with further energy costs.

Response: Following the reviewer's suggestion, we have removed "pressurization" from Scheme 1 to avoid being misleading. Our high-pressure CO₂R system takes the advantages of industrial CO₂ or captured CO₂ that are already pressurized in upstream processes.

- A local pH of 12.3 is reported at high pressure, and it is claimed that it is higher compared to the case of atmospheric pressure. Is this really the case, if we apply the same current density? This should be supported by experimental results.

Response: We agree with the reviewer that the local pH in these two cases will be close when we apply the same current density, even though the production of one OH⁻ corresponds to the transfer of two electrons for CO₂ to formate conversion and one electron for HER.

However, the CO₂R product selectivity is strongly related to the cathode potential, we therefore applied the same potential (-1.1 V vs. RHE) under both atmospheric and 50 bar conditions for fair comparison. If applying the same current density, the atmospheric pressure scenario would induce more OH⁻ production compared to the high-pressure scenario, because of the higher HER selectivity (CO₂ + H₂O + 2e⁻ → HCOO⁻ + OH⁻ vs. 2H₂O + 2e⁻ → H₂ + 2OH⁻). As a result, the cathode surface pH under

atmospheric pressure will be slightly higher than that under high pressure. However, the difference of local OH⁻ concentrations near the cathode between the two scenarios is very small (< 5 mM at 80 mA cm⁻²). It is beyond the capability of current Raman systems to differentiate such a small pH difference ($\Delta \text{pH} < 0.2$). Therefore, we have revised the text to remove this claim, and we now indicate only that we expect a small difference in pH due to the factors noted above.

- The formation of OH⁻ is said to be a parasitic process. It is not parasitic, it is part of the stoichiometry of CO₂ reduction.

Response: As suggested, we have corrected the description in the revised manuscript.

- The theory of needing a hydronium-repellent layer to increase formate selectivity seems a bit bizarre, as Cu catalysts are typically used in 1M (or more concentrated) alkaline electrolyte solutions, where very little formation production is observed. Considering this, I do not really see the relevance of the DFT studies where hydronium ions are brought to the surface.

Response: We agree with the reviewer that hydronium ions cannot be considered as the main factor influencing formate selectivity under alkaline conditions. Therefore, we rebuilt the theoretical model using CO₂ coverage as the primary factor for the two CO₂R pathways and HER. As shown in Fig. R1, the increase in CO₂ coverage greatly increased the free energy change of the potential-determining step (PDS) for *CO production (PDS_{CO}), while decreased the free energy change of PDS for HCOOH (PDS_{HCOOH}). Additionally, we observed that the variation in CO₂ coverage exerted insignificant impact on the adsorption of *H – the PDS for HER, consistent with the experimental results.

Fig. R1 Free energy change of CO₂R with various CO₂ coverage. (a) Free energy diagram from CO₂ toward *CO. (b) Free energy diagram from CO₂ toward HCOOH. (c) Free energy changes of PDS in CO₂R for *CO or HCOOH production on Cu(111) with various CO₂ coverages. (d) Free energy diagram of HER on Cu(111) with various CO₂ coverages.

The new data was added in Fig. 3 and the analysis was highlighted in the manuscript as follows:

These findings motivated us to investigate the CO₂R reaction mechanism at high pressure using DFT. In CO₂R, oxygen or carbon atoms of CO₂ can be protonated to *COOH or *HCOO, respectively – the key

intermediates for CO₂R in branching to *CO vs. HCOOH pathways (Supplementary Fig. 14)⁴⁸. In light of the pressure-dependent CO₂ solubility and local microenvironment near the cathode surface (Fig. 2c), we sought to explore the impact of CO₂ coverage on CO₂R on a Cu(111) facet – the dominant facet of the as-prepared Cu catalyst (Supplementary Fig. 1f). As depicted in Fig. 3a and 3b, the calculated energy diagrams on the periodic Cu(111) surface indicate the potential-determining steps (PDS) for two CO₂R pathways – the formation of *COOH for *CO pathway (PDS_{CO}) and the hydrogenation of *HCOO for HCOOH pathway (PDS_{HCOOH}), as seen in previous reports⁴⁹. The free energies of forming *COOH and *HCOO both increased with the CO₂ coverage varying from 1/9 monolayer (ML) to 3/9 ML. Consequently, the free energy change of PDS_{CO} increased, while that of PDS_{HCOOH} decreased – that means, with higher CO₂ coverages, *CO production became more difficult whereas HCOOH production became more energetically favorable (Fig. 3c), leading to the CO₂R product selectivity shift toward formate/formic acid. It should be noted that due to the imprecise portrayal of carbon-oxygen double bonds in DFT, we focused on the variation trends of the free energies, instead of their absolute values⁵⁰. We then studied the effect of CO₂ coverage on the side reaction – the hydrogen evolution reaction (HER; Supplementary Fig. 15). The free energy diagram was calculated based on the Langmuir-Hinshelwood (LH)-type mechanism, and the adsorption of *H was identified as the PDS for HER (PDS_{H2}). As the CO₂ coverage increased from 0 to 3/9 ML, the adsorption strength of *H was weakened insignificantly with a decrease < 0.05 eV (Fig. 3d). This finding is in line with the experimental observation of the negligible decrease in H₂ selectivity when elevating the pressure (Supplementary Fig. 16).

- PPy is deposited oxidatively. This can alter the surface of the catalyst. This must be addressed.

Response: According to the reviewer's suggestion, we conducted supplementary experiments and theoretical simulations to demonstrate the potential influence of PPy on the Cu surface and its CO₂R performance.

The Cu/PPy catalyst was synthesized by electro-oxidative polymerization of pyrrole monomers to form a polymeric layer on the Cu(OH)₂ surface (Cu(OH)₂/PPy), which was then reduced to Cu/PPy at a reduction potential (Supplementary Fig. 17a). The Cu(OH)₂ precursor was kept in a fully oxidized state prior to the deposition of PPy and thus remained stable throughout the synthesis process. We then peeled off the Cu/PPy nanoparticles from the substrate for comparison with the naked Cu catalyst. XRD results showed that they both exhibited a polycrystalline Cu structure with similar grain size, intensity, and crystallinity (Fig. R2a). Additionally, high-resolution TEM images showed no special crystal plane or defect on the surface of Cu/PPy (Fig. R2b). Therefore, we concluded that PPy did not pose significant impact on the morphology and structure of the Cu catalyst. In the revised manuscript, the new HRTEM image was added as Supplementary Fig. 18b.

Fig. R2 (a) XRD patterns of Cu and Cu/PPy nanoparticles. The full width at half maximum (FWHM) and crystalline size (CS) were tested and calculated under the same conditions. (b) TEM image of the Cu/PPy nanoparticles. The inset was the corresponding fast Fourier transform (FFT) pattern.

- PPy might bind CO₂, or some reaction intermediates, that can alter the course of the reaction.

Response: On the one hand, PPy is a linear or mesh-like polymer film formed through electro-oxidative polymerization of pyrrole monomers and it loosely enwrap the Cu surface (Fig. R3a). Therefore, PPy could barely influence the adsorption/desorption of CO₂ molecules or CO₂R intermediates on the Cu surface. Moreover, PPy on its own exhibited almost no CO₂R activity, and the CO₂R product distribution of Cu/PPy was consistent with that of the Cu catalyst. This indicated that the Cu surface remained as the main active sites in Cu/PPy.

On the other hand, the DFT calculation suggested that PPy possessed a coordination-saturated structure, and that the van der Waals interaction between the non-metallic atoms and intermediates such as *CO was weak – that means, PPy cannot bind or adsorb the CO₂R intermediates (Fig. R3b). Based on these results, we concluded that PPy improved the CO₂R performance of Cu under high pressure by suppressing the HER, instead of posing impact on the adsorption of CO₂ or reaction intermediates (*Adv. Funct. Mater.* **2014**, *24*, 1265-1274; *J. Catal.* **2021**, *393*, 92-99). In the revised manuscript, we have presented these new data in Supplementary Fig. 18.

Fig. R3 (a) STEM image of the Cu/PPy catalyst. The inset showed the simulated interlayer spacing between Cu(111) facet and PPy layer. The red arrows indicated the areas with loose PPy or naked Cu sites. (b) Optimized structure of adsorbed intermediates on the PPy layer. Cu: orange; N: blue; O: red; C: gray; H: white.

- Coating the catalyst with PPy might alter its electronic structure, that can also change the reaction selectivity.

Response: We evaluated the effect of PPy on the electronic structure of the Cu surface through XPS and DFT. By comparing the characteristic peaks and valence states of Cu in XPS spectra (Fig. R4a), we found that there was no significant shift in the peak position between the Cu and Cu/PPy catalysts. In addition, the charge density difference calculated by DFT showed that PPy and Cu interacted weakly by van der Waals forces (Fig. R4b), and their atomic distance was large (3.2 Å) – that means weak electronic interaction. Therefore, the influence of PPy layer on the electronic structure of Cu surface was negligible. In the revised manuscript, we have presented these new data in Supplementary Fig. 18.

Fig. R4 (a) Cu 2p XPS spectra of the Cu and Cu/PPy catalysts. (b) The charge density difference for the Cu/PPy catalyst. The yellow and cyan colors indicated the charge accumulation and depletion, respectively. Cu: orange; N: blue; O: red; C: gray; H: white.

- Was the current density identical when comparing the surface pH of the bare and PPy coated catalysts?

Response: The CO₂R product selectivity is strongly related to the cathode potential. Therefore, we controlled the same cathode potential on the two catalysts for fair comparison, and the current densities were comparable (192 mA cm⁻² on Cu vs. 229 mA cm⁻² on Cu/PPy) but not identical.

- HCOO⁻ relative purity is shown in Fig 5. I understand that this is related to the liquid products, but I still find it misleading. The formate concentration should be presented.

Response: Following the reviewer's suggestion, we have removed the formate relative purity curve from Fig. 5 and presented the formate concentration over time in Supplementary Fig. 22b (Fig. R5).

Fig. R5 Formate concentration and relative purity over 12 h continuous CO₂R in the narrow-gap aqueous flow cell at a current density of 400 mA cm⁻² under 50 bar.

Reviewer #2 (Remarks to the Author):

Summary of the paper: The authors investigate how pressurization affects the CO₂RR. To this end, they developed new operando IR experimental techniques. This allows to study the electrochemical reaction under pressure. They find that higher pressure alters the mechanism to favor formate, most likely due to a different ratio of CO₂* and hydronium ions. They employ both computational and experimental techniques to elucidate the different product selectivity and rationally improve the catalyst based on their mechanistic findings.

• Significance:

Their significance for the catalysis community is very high due to several reasons:

First, the authors develop new operando IR techniques which can provide crucial mechanistic insights for pressurized electrochemical experiments. This type of method development is crucial for obtaining insights in catalytic mechanisms.

Second, this study is a great example of how the combination of experimental and computational insights can be utilized to rationally improve the catalyst.

Response: We are very grateful to the reviewer for her/his appreciation of our work. Following the reviewer's insightful and constructive comments, we have performed new experiments and theoretical simulations, and carefully revised our manuscript.

• Unfortunately, I don't believe that this study is suitable for a broad audience such as Nature Communications. The reasons are outlined below (a-c):

Response: We appreciate the reviewer for pointing out the shortcomings in our work, but we respectfully disagree that our findings would not attract a broad audience. Currently, CO₂R is one of the most popular research topics. However, the field has not paid enough attention to pressure – a very important factor in industry. Our work highlighted the potential of leveraging high-pressure CO₂ for electrochemical production of valuables, which would interest not only researchers in the electrocatalysis field, but also relevant industrial stakeholders. We will provide more detailed explanations to the three specific points raised by the reviewer as follows:

a) First, optimizing reaction conditions to yield 2 electron products such as CO or CHOOH has been studied in great detail in the literature. Altering the product distribution to yield four or six electrons such as C₂ products would be highly desirable.

Response: We agree with the reviewer that electrochemically synthesizing multi-carbon (C₂₊) products would be highly desirable, but this does not necessarily mean that HCOOH or CO are of lower value or importance. As highlighted in a techno-economic analysis by the Jiao group (*Ind. Eng. Chem. Res.* **2018**, *57*, 2165–2177), HCOOH and CO are still the only economically viable CO₂R products. This is due to their generally higher Faradic efficiency (FE) and energy efficiency (EE) over C₂₊ products.

More importantly, our work focused on studying the pressure-mediated CO₂R mechanism and demonstrating the cathode design principles for high-pressure scenarios, instead of the product itself.

b) Second, while the product distribution of the CO₂RR changes using pressure the selectivity of CO₂RR versus HER is not strongly affected as seen in Fig. 1b.

Response: Accordingly, we rebuilt the DFT model using CO₂ coverage as the primary factor and investigated its effect on HER (Fig. R6a). As shown in Fig. R6b, the *H adsorption was the potential-determining step (PDS) for HER. When the CO₂ coverage increased, the Gibbs free energy of *H on the Cu surface slightly changed for < 0.05 eV, indicating that the pressure posed insignificant impact on the *H adsorption strength (Fig. R6b). These results were consistent with our experimental observations that HER was not strongly affected by the pressure (Fig. R6c).

Fig. R6 (a) Optimized structures of *H for HER with various CO₂ coverages. Cu: orange; O: red; C: gray; H: white. (b) Free energy change of HER on Cu(111) with various CO₂ coverages. (c) FE of H₂ on the Cu catalyst under different pressures at -1.1 V vs. RHE.

We have added these new data in Fig. 3, Supplementary Figs. 15 and 16, and highlighted the analysis in the revised manuscript as follows:

We then studied the effect of CO₂ coverage on the side reaction – the hydrogen evolution reaction (HER; Supplementary Fig. 15). The free energy diagram was calculated based on the Langmuir-Hinshelwood (LH)-type mechanism, and the adsorption of *H was identified as the PDS for HER (PDS_{H₂}). As the CO₂ coverage increased from 0 to 3/9 ML, the adsorption strength of *H was weakened insignificantly with a decrease < 0.05 eV (Fig. 3d). This finding is in line with the experimental observation of the negligible decrease in H₂ selectivity when elevating the pressure (Supplementary Fig. 16).

c) Third, the effect that pCO₂RR changes the product distribution has been observed before (as acknowledge by the authors).

Response: As pointed out by reviewer, there were indeed some reports that observed the CO₂R product selectivity shift under pressure. However, the underlying mechanism remain unexplored. In this study, we reveal the pressure-dependent CO₂R mechanism by the close interplay of electrochemical measurements, theoretical calculations, and quantitative *operando* Raman spectroscopy that can depict the local microenvironment near the cathode surface under high pressure. Furthermore, we demonstrated how the mechanism could guide us to rationally retrofit the non-selective Cu electrode for a higher formate productivity. Therefore, our work not only provided new insights in high-pressure CO₂R, but also showcased a frontier cathode design method.

• Major points:

The authors developed new experimental techniques which give crucial mechanistic insights. This guided a well-designed DFT study. In turn, those insights are indirectly confirmed by rational modification to the catalyst by introducing PPy further improving the performance. I really enjoyed reading this manuscript.

a) One concern I have regarding the DFT simulation are the ddG plots in Fig 3 a and b. The predicted results do not follow the experimental FE in 1b. The ddG changes only slightly as function of CO₂* coverage and even at 1/9 the difference is larger than 0.5 eV suggesting that formic acid should already be the major product. The author should add clarifying statements about expected accuracies of the used DFT framework and cite similar studies. In addition, it should not be presented as a driving indicator for the change in product distribution.

Response: As suggested by the reviewer, we rebuilt the theoretical model using CO₂ coverage as the primary factor for the CO₂R pathways towards *CO and HCOOH (Fig. R7). Specifically, we calculated the free energy diagrams and identified the PDS – the formation of *COOH for *CO pathway (PDS_{CO}) (Fig. R8a) and the hydrogenation of *HCOO for HCOOH pathway (Fig. R8b). We have also observed that with higher CO₂ coverages, the adsorption strengths of both *COOH and *HCOO weakened. Consequently, the free energy change of PDS_{CO} increased, while that of PDS_{HCOOH} decreased – that means, *CO production became more difficult whereas HCOOH production became more energetically favorable (Fig. R8c). These findings suggested that CO₂ coverage played an important role in the shift of CO₂R product selectivity toward formate/formic acid under pressure.

Fig. R7 The optimized structures of adsorbed CO₂R intermediates for HCOOH and *CO production with various CO₂ coverages. Cu: orange; O: red; C: gray; H: white. The water molecules in the solvation were represented using stick models to clearly show the adsorbed intermediates on the (3x3) Cu(111) surface.

Fig. R8 Free energy diagram from CO₂ toward (a) *CO and (b) HCOOH. (c) Free energy changes of

PDS in CO₂R for *CO or HCOOH production on Cu(111) with various CO₂ coverages.

We have added these new data in Fig. 3 and Supplementary Fig. 14, and highlighted the analysis in the revised manuscript as follows:

These findings motivated us to investigate the CO₂R reaction mechanism at high pressure using DFT. In CO₂R, oxygen or carbon atoms of CO₂ can be protonated to *COOH or *HCOO, respectively – the key intermediates for CO₂R in branching to *CO vs. HCOOH pathways (Supplementary Fig. 14)⁴⁸. In light of the pressure-dependent CO₂ solubility and local microenvironment near the cathode surface (Fig. 2c), we sought to explore the impact of CO₂ coverage on CO₂R on a Cu(111) facet – the dominant facet of the as-prepared Cu catalyst (Supplementary Fig. 1f). As depicted in Fig. 3a and 3b, the calculated energy diagrams on the periodic Cu(111) surface indicate the potential-determining steps (PDS) for two CO₂R pathways – the formation of *COOH for *CO pathway (PDS_{CO}) and the hydrogenation of *HCOO for HCOOH pathway (PDS_{HCOOH}), as seen in previous reports⁴⁹. The free energies of forming *COOH and *HCOO both increased with the CO₂ coverage varying from 1/9 monolayer (ML) to 3/9 ML. Consequently, the free energy change of PDS_{CO} increased, while that of PDS_{HCOOH} decreased – that means, with higher CO₂ coverages, *CO production became more difficult whereas HCOOH production became more energetically favorable (Fig. 3c), leading to the CO₂R product selectivity shift toward formate/formic acid.

Theoretical methods

All density functional theory (DFT) calculations were performed by the Vienna ab initio simulation program (VASP)^{55,56}. The core-valence interactions were calculated by the project augmented wave (PAW) method with 450 eV as the cut-off energy^{57,58}. The generalized gradient approximation in the Perdew–Burke–Ernzerhof functional (GGA-PBE) was applied to describe the exchange-correlation correction effect⁵⁹. The DFT-D3 method was used to consider the dispersion correction of the van der Waals force⁶⁰. For the geometry optimization, the self-consistent iteration must reach 10⁻⁶ eV for the energy convergence and 0.01 eV Å⁻¹ for the force convergence.

The (3 × 3) Cu(111) model consisted of four Cu atomic layers, where the two bottom layers were fixed to mimic the bulk material and all other degrees of freedom were relaxed. To account for both field and solvation effects, we incorporated one charged layer of water molecules onto the Cu(111) surface at the intermediates⁶¹. The vacuum space was set at about 15 Å in z-axis to avoid interactions between the periodic images. To investigate the impact of CO₂ coverage, we conducted a comprehensive analysis involving the adsorption of one, two, or three CO₂ molecules onto the periodic (3 × 3) Cu(111) surface. Two key reaction pathways of CO₂R toward *CO or HCOOH were considered as follows:

CO₂ to HCOOH:

CO₂ to *CO:

Additionally, the elementary steps of HER were listed as below:

where * represents the active site or the adsorbed intermediate. The proton-electron pair was treated with the computational hydrogen electrode (CHE) model⁴⁹. The PDS in the three reaction pathways was identified based on the most positive change in free energy. A more positive change in PDS indicated a more difficult reaction pathway. The optimized structures for all reactions were presented in the Supplementary Figs. 14 and 15. The free energy change of each elementary step for the production of *CO, HCOOH_(l), or H_{2(g)} was obtained by the correction of Gibbs free energy (*G*) at room temperature (T = 298.15 K), using the following equation:

$$G = E_{DFT} + ZPE + \int C_p dT - TS$$

where *E*_{DFT} is the energy changes of DFT calculations, *ZPE* is the zero-point energy, *C_p* is the heat capacity, and *S* is the entropy.

b) Another interesting point omitted in their simulations is HER side reaction. If the hydronium ions dictate the product distribution of the CO₂RR, a similar effect should be observed for HER activity. It seems not intuitive to me that this is not the case. Simulation should be carried out to investigate this (perhaps in a separate study).

Response: We agree with the reviewer that hydronium ions cannot be considered as the main factor. Therefore, we rebuilt the theoretical model using CO₂ coverage as the primary factor and studied the effect of CO₂ coverage on HER. The free energy diagram was calculated based on the Langmuir-Hinshelwood (LH)-type mechanism (Fig. R6a), and the adsorption of *H was identified as the PDS for HER (PDS_{H₂}). As the CO₂ coverage increased from 0 to 3/9 ML, the adsorption strength of *H was weakened insignificantly with a decrease < 0.05 eV (Fig. R6b). This finding was in line with our experimental observation of the negligible decrease in H₂ selectivity when elevating the pressure (Fig. R6c).

The new data were added in Fig. 3d, and the analysis was highlighted in the revised manuscript as follows:

We then studied the effect of CO₂ coverage on the side reaction – the hydrogen evolution reaction (HER; Supplementary Fig. 15). The free energy diagram was calculated based on the Langmuir-Hinshelwood (LH)-type mechanism, and the adsorption of *H was identified as the PDS for HER (PDS_{H₂}). As the CO₂ coverage increased from 0 to 3/9 ML, the adsorption strength of *H was weakened insignificantly with a decrease < 0.05 eV (Fig. 3d). This finding is in line with the experimental observation of the negligible decrease in H₂ selectivity when elevating the pressure (Supplementary Fig. 16).

• **Minor points:**

a) The choice of 6 water molecules seems a bit arbitrary, the authors should justify this choice and show how their findings converge with respect to the number of water molecules. The same should be done for adding hydronium ions, it is not clear how many were added and several should be tested and reported.

Response: We acknowledge that the previous description of the DFT model was not precise enough and should be clarified. Specifically, we would like to emphasize that the ring configuration of the water hexamer (Fig. R9) on Cu(111) was composed of 6 water molecules, as determined by a combination of theoretical studies and scanning tunneling microscopy (STM) characterizations (*J. Chem. Phys.* **2002**, *116*, 5746-5752; *Nat. Mater.* **2007**, *6*, 597-601; *J. Am. Chem. Soc.* **2020**, *142*, 6902–6906). To model the solvation effect, we added proton bonding with a water molecule to create H₃O⁺, based on the research conducted by Nørskov et al (*J. Phys. Chem. Lett.* **2015**, *6*, 2032–2037). Our model involved a charged water layer on the periodic Cu(111)-(3x3) surface, which allowed us to consider the combination of field effects and solvation effects. This model with explicit solvation has demonstrated its rationality and scientific validity, and has been widely used in previous theoretical calculations for CO₂R (*Nat. Commun.* **2017**, *8*, 15438; *Nat. Catal.* **2019**, *2*, 1124-1131). The number of hydronium ions does not need to be discussed since we rebuilt the theoretical model using CO₂ coverage as the primary factor.

Fig. R9 The periodic (3x3) Cu(111) surface with a charged water layer. Cu: orange; O: red; C: gray; H: white.

b) The axis labels and ticks could be larger in the images

Response: Following the reviewer's suggestion, we have enlarged the axis labels and ticks accordingly.

c) Fig 4b does not give a lot of insight and can be moved into the SI

Response: As suggested by the reviewer, we have moved Fig. 4b to Supplementary Fig. 17d.

• **Scholar representation:**

Well written, very clear figures and amazing illustrations. The authors should cite more work about the pressurized CO₂RR such as the original report from the 1990s.

Response: Following the reviewer's suggestion, we have carefully revised the references (Refs. 20-33), added and highlighted the new discussions in the revised manuscript as follows:

In prior studies that sought to leverage pressure in aqueous-based CO₂R^{20,21}, PCO₂ was reduced to CO or formate²². These include examinations of altered CO₂R product selectivity on various metal catalysts under high pressure²³⁻²⁶. A Ni wire electrode that had no CO₂R activity under ambient pressure showed 23% formic acid selectivity under 60 bar²⁵. Enhanced formate selectivity was seen on Sn using PCO₂^{27,28}. Theoretical modelling and control experiments were also conducted to understand CO₂R under high pressure²⁹⁻³³. More recently, PCO₂ has been found to transform Cu-based catalysts to become formate-selective³⁴. Topics of interest remain to be studied in depth, such as the detailed influence of pressure on the local microenvironment near the CO₂R electrode.

Reviewer #3 (Remarks to the Author):

In the article " Pressure dependence in aqueous-based electrochemical CO₂ reduction", the authors have discussed the effect of the pressure on the CO₂ electrochemical process on Cu, Ag, Au and Sn cathode. Electrochemical reduction of CO₂ is among the most successful methods of conversion of this waste gas to high value-added products and the process is under intense investigation by many researchers. The manuscript is well-written and concise. Though the results are very interesting some questions/concerns need to be addressed before the final acceptance.

Response: We are very grateful to the reviewer for her/his appreciation of our work. Following the reviewer's insightful and constructive comments, we have made thorough revisions to the manuscripts, including the addition of citations and discussions.

- This work suffers from a lack of previous important achievements reported in the literature by several authors on the electrochemical CO₂ conversion under high pressure of CO₂. First of all, the authors report that they "discover that pressurization up to 50 bar regulates common CO₂R catalysts to be formate selective in aqueous systems", however, the effect of the CO₂ pressure on the selectivity of the process as well as on the mechanism reaction under high pressure was already reported. In 1993, Kudo et al. (J. Electrochem. Soc. 140 (1993) 1541, doi:10.1149/1.2221599) showed that an increase of CO₂ pressure allowed to favor CO₂ reduction to formic acid using a Ni wire electrode. Similar results were reported by Hara et al. (J. Electroanal. Chem. 391 (1995) 141–147. doi:10.1016/0022-0728(95)03935-A.) and Mizuno et al. (Energy Sources 17 (1995) 503–508. doi:10.1080/00908312.). As well as interesting results on the reaction mechanism of the CO₂ conversion under high pressure were found for example by Proietto et al. (ChemElectroChem 6 (2019) 162–172, doi:10.1002/celec.201801067) and Morrison et al. (J. Electrochem. Soc. 166 (2019) E77–E86. doi:10.1149/2.0121904jes). Other citations in this field are missing; see for examples: Fresenius Environmental Bull 12 (2003) 1202–1206; Appl. Catal. A Gen. 274 (2004) 237–242; Journal of The Electrochemical Society, 159 (9) (2012) F514-F517; Energy Environ. Sci., 11 (2018) 2531; Ind. Eng. Chem. Res. 58 (2019) 1834–1847; Electrochim. Acta 199 (2015) 332–341; Journal of CO₂ Utilization 67 (2023) 102338; and so on. Hence, the authors should consider improving the introduction section and to discuss the results according to the literature.

Response: Following the reviewer's suggestion, we have more comprehensively referenced this literature (Refs. 20-33) and added discussions to the introduction section accordingly:

In prior studies that sought to lever pressure in aqueous-based CO₂R^{20,21}, PCO₂ was reduced to CO or formate²². These includes examinations of altered CO₂R product selectivity on various metal catalysts under high pressure²³⁻²⁶. A Ni wire electrode that had no CO₂R activity under ambient pressure showed 23% formic acid selectivity under 60 bar²⁵. Enhanced formate selectivity was seen on Sn using PCO₂^{27,28}. Theoretical modelling and control experiments were also conducted to understand CO₂R under high pressure²⁹⁻³³. More recently, PCO₂ has been found to transform Cu-based catalysts to become formate-selective³⁴. Topics of interest remain to be studied in depth, sch as the detailed influence of pressure on the local microenvironment near the CO₂R electrode.

- Please, consider that reference 13 seems not appropriate to the context in the following sentence: "However, industry works exclusively with pressurized CO₂ (PCO₂) in capture, transport and storage. In many such cases, the CO₂ is in dissolved form (1 - 110 bar)13,14." Hence, I would suggest clarifying this content or removing the citation.

Response: Following the reviewer's suggestion, we have removed reference 13.

- The authors should explain why they used 1 bar of CO₂ and 49 bar of Ar. Why the choice to investigate the process under a final pressure of 50 bar?

Response: When using 50 bar CO₂, there were two factors that may post impact on CO₂R: 1) the high pressure; and 2) the high CO₂ solubility. To exclusively investigate the influence of pressure, rather than the coeffects of pressure and CO₂ solubility, we used the mixed gas of 1 bar CO₂ and 49 bar Ar to control

the CO₂ solubility the same as that under ambient pressure.

Regarding the final pressure, we chose 50 bar because 1) the CO₂ solubility significantly increases when elevating the pressure up to 50 bar (*Fluid Phase Equilibria* **2003**, 208, 265-290); however, the increase in CO₂ solubility decelerates under higher pressures; 2) under pressure > 50 bar, CO₂ may encounter a phase transition and be transformed into liquid (> 56.7 bar, 20 °C) or supercritical (> 73.2 bar, 31.1 °C) state; and 3) experimental results indicated that further elevating the pressure beyond 50 bar can limitedly increase the CO₂ coverage and formate productivity, as shown in Supplementary Fig. 16. Therefore, we chose 50 bar as the final pressure.

- One of the main technological problems on the electrochemical cell under high pressure is the leakage of gas; the author should give more information of the high-pressure narrow-gap aqueous flow cell. It is not possible to understand how the pressure balance was assured between the two compartments and how you can reproduce these experiments.

Response: As suggested by the reviewer, we have added a flow diagram (Supplementary Fig. 21a; also presented in Fig. R10) and detailed description of the high-pressure narrow-gap aqueous flow cell in the revised manuscript, including the experimental setup, individual components, key parameters, and operation procedures:

High-pressure narrow-gap aqueous flow cell

The high-pressure narrow-gap aqueous flow cell system (Supplementary Fig. 21a) consisted primarily of a narrow-gap aqueous flow cell, two high-pressure high-performance liquid chromatography pumps (HPLC pump; Sanotac SP6010), two backpressure valves (Beijing Xiong Chuan Technology Co. LTD), two Teflon-lined titanium tanks containing 1.0 M KHCO₃ catholyte and 0.5 M K₂SO₄ anolyte, respectively, and a CO₂ gas cylinder. The high-pressure narrow-gap aqueous flow cell was assembled by stacking the following components in order: a Cu/PPy cathode sandwiched by two polytetrafluoroethylene (PTFE) gaskets (0.03 cm thick with a 0.5 cm × 1 cm window as the reactive area) as the catholyte compartments, a Nafion 117 membrane, and a RuO₂/Ti foam anode sandwiched by two aforementioned PTFE gaskets as the anolyte compartments (Fig. 5a and Supplementary Fig. 21b). These components were fixed and sealed by two titanium plates with channels and ports. The catholyte and anolyte were pressurized by high-pressure HPLC pumps and equilibrated by backpressure valves. Prior to each experiment, the air in the catholyte was purged out by bubbling CO₂ under atmospheric pressure. Then, the pressures of CO₂ and electrolytes were simultaneously and gradually increased by adjusting the gas cylinder valve and backpressure valves. Both cathode and anode compartments of the narrow-gap aqueous flow cell were pressurized to 50 bar and held for 30 min to achieve the equilibrium solubility of CO₂ in catholyte. During CO₂R, the electrolytes were recirculated at a constant flow rate of 10 mL min⁻¹. The anodic O₂ and cathodic CO₂/CO₂R products were discharged separately through the outlets of each backpressure valves. The CO₂R gas products were collected by a syringe and analyzed by GC, while the liquid products were sampled by withdrawing the catholyte solution through the sampling port every hour and analyzed by NMR.

Fig. R10 Flow diagram of the high-pressure narrow-gap aqueous flow cell system.

REVIEWER COMMENTS

Reviewer #1 (Remarks to the Author):

The authors revised the manuscript carefully, and I am satisfied with the responses they provided to the raised points/questions. Therefore I suggest the publication of this manuscript.

Reviewer #2 (Remarks to the Author):

Summary of the paper: The authors investigate how pressurization affects the CO₂RR. To this end, they developed new operando IR experimental techniques. This allows to study the electrochemical reaction under pressure. They find that higher pressure alters the mechanism to favor formate, most likely due to a different ratio of CO₂* and hydronium ions. They employ both computational and experimental techniques to elucidate the different product selectivity and rationally improve the catalyst based on their mechanistic findings.

The revised manuscript addresses most of my initial concerns and their improved manuscript is almost ready for publication. A few minor points should be addressed:

The citations of the original work has been improved now, the introduction should also include shortcoming of the most important studies briefly laid out to help motivate the impact of this work.

The new model is convincing and inline with experimental results. However, a little more insight from these DFT calculations could be presented to better rationalize the pressure trends in HER and the two CO₂RR pathways.

Minor points:

The color palette choice in Figs 4 a, b, d is a little unclear. It is hard to distinguish the lighter colors at first glance.

Supp. Figs. 14 and 15 should have a better perspective; maybe the same perspective as original Fig. 3c from main manuscript

The response to minor point a): The choice of 6 water molecules seems a bit arbitrary, the authors should justify this choice ... should be added in a condensed form to the methods Theoretical methods section of the main manuscript

Reviewer #3 (Remarks to the Author):

The manuscript has been improved according to all the indications of the reviewers. Hence, it can be published without further revisions.

Responses to Reviewers

Reviewer #1 (Remarks to the Author):

The authors revised the manuscript carefully, and I am satisfied with the responses they provided to the raised points/questions. Therefore I suggest the publication of this manuscript.

Response: We would like to express our gratitude to the Reviewers for their time and efforts in reviewing our manuscript. Their comments have greatly assisted us in improving our manuscript.

Reviewer #2 (Remarks to the Author):

Summary of the paper: The authors investigate how pressurization affects the CO₂RR. To this end, they developed new operando IR experimental techniques. This allows to study the electrochemical reaction under pressure. They find that higher pressure alters the mechanism to favor formate, most likely due to a different ratio of CO₂* and hydronium ions. They employ both computational and experimental techniques to elucidate the different product selectivity and rationally improve the catalyst based on their mechanistic findings.

The revised manuscript addresses most of my initial concerns and their improved manuscript is almost ready for publication. A few minor points should be addressed:

Response: Following the reviewer's suggestions, we have augmented the Introduction and DFT sections with supplementary discussions, and improved the figure colors and perspectives.

- The citations of the original work has been improved now, the introduction should also include shortcoming of the most important studies briefly laid out to help motivate the impact of this work.

Response: As suggested, we have included more discussions on the shortcomings in reported works to highlight the impact of our work. The added part was highlighted in the Introduction section as follows:

“Although these results have shown the impact of pressure on CO₂R, the underlying mechanism of the pressure-dependent CO₂R selectivity has yet to be systemically revealed. In particular, the local microenvironment near the CO₂R electrode (such as the concentrations of key species, pHs, etc.) under the influence of pressure is critical to the final CO₂R pathway, but has been rarely observed. This task is beyond...”

- The new model is convincing and inline with experimental results. However, a little more insight from these DFT calculations could be presented to better rationalize the pressure trends in HER and the two CO₂RR pathways.

Response: Following the reviewer's suggestion, we have thoroughly discussed the trends in the activity of HER and the pathways of CO₂R under different CO₂ coverages according to the DFT calculations. The added part was highlighted in the manuscript as follows:

“However, with higher CO₂ coverages, the free energy change (ΔG) of PDS_{CO} increased, while that of PDS_{HCOOH} decreased. Specifically, at a CO₂ coverage of 1/9 ML, the ΔG of PDS_{CO} was notably lower than that of PDS_{HCOOH}, indicating that the *CO pathway was dominant. The situation was reversed when the CO₂ coverage gradually increased to 3/9 ML, where the ΔG of PDS_{CO} increased to 1.04 eV and that of PDS_{HCOOH} decreased to 0.75 eV – that means, the *CO pathway became more difficult whereas HCOOH production became more energetically favorable (Fig. 3c). The DFT models revealed that the pressure-dependent CO₂ coverage played a crucial role in shifting the CO₂R product selectivity towards formate/formic acid... We then studied the effect of CO₂ coverage on the side reaction – the hydrogen evolution reaction (HER)... As the CO₂ coverage increased from 0 to 2/9 ML, the adsorption strength of *H was slightly decreased, but no further decrease was observed at a higher CO₂ coverage of 3/9 ML (Fig. 3d). The insignificant ΔG of PDS_{H₂} suggested the weak impact of CO₂ coverages on HER, which was consistent with the experimental observation of the slight decrease in H₂ selectivity under elevated pressures.”

Minor points:

- The color palette choice in Figs 3 a, b, d is a little unclear. It is hard to distinguish the lighter colors at first glance.

Response: Following the reviewer's suggestion, we have improved the curve colors in Fig. 3 for better contrast.

- Supp. Figs. 14 and 15 should have a better perspective; maybe the same perspective as original Fig. 3c from main manuscript.

Response: Following the reviewer's suggestion, we have replotted and added the side views of the DFT models in Supplementary Figs. 14 and 15 for better perspectives of the model configurations.

- The response to minor point a): The choice of 6 water molecules seems a bit arbitrary, the authors should justify this choice ... should be added in a condensed form to the methods Theoretical methods section of the main manuscript.

Response: As suggested, we have added the methodology and description of constructing one charged water layer on Cu(111) model to the Theoretical Methods section. The description was highlighted in the manuscript as follows:

“To account for both explicit solvation and field effects, we incorporated one charged water layer onto the Cu(111) surface at the intermediates according to studies of Nørskov et al⁶¹, where the optimal water structure are obtained via using a minima-hopping algorithm⁴⁹. Here, the periodic structure of one charged water layer, consisting of five water molecules and one hydronium molecule in the (3 × 3) cell, closely resembles the hexagonal ice-like structure using previously in various DFT-based studies of adsorption and proton-coupled electron transfer (PCET) kinetics on Pt(111)^{62, 63}, and has been widely used in the CO₂R studies⁶⁴.”

Reviewer #3 (Remarks to the Author):

The manuscript has been improved according to all the indications of the reviewers. Hence, it can be published without further revisions.

Response: We would like to once again thank the Reviewers very much for their constructive suggestions.